# NEURAL STRUCTURE MAPPING FOR LEARNING ABSTRACT VISUAL ANALOGIES

## ABSTRACT

Building conceptual abstractions from sensory information and then reasoning about them is central to human intelligence. Abstract reasoning both relies on, and is facilitated by, our ability to make analogies about concepts from known domains to novel domains. Structure Mapping Theory of human analogical reasoning posits that analogical mappings rely on (higher-order) relations and not on the sensory content of the domain. This enables humans to reason systematically about novel domains, a problem with which machine learning (ML) models tend to struggle. We introduce a two-stage neural framework, which we call Neural Structure Mapping (NSM), to learn visual analogies from Raven's Progressive Matrices, an abstract visual reasoning test of fluid intelligence. Our framework uses (1) a multi-task visual relationship encoder to extract constituent concepts from raw visual input in the source domain, and (2) a neural module net-based analogy inference engine to reason compositionally about the inferred relation in the target domain. Our NSM approach (a) isolates the relational structure from the source domain with high accuracy, and (b) successfully utilizes this structure for analogical reasoning in the target domain.

## 1 INTRODUCTION

The ability to form abstractions of 'concepts' from information and then reason about them is central to human intelligence (Lake et al., 2015). Abstractions enable humans to quickly learn concepts from few examples, and then reason systematically about new concepts by composing previously understood concepts (Nam & McClelland, 2021). Over the last decade, artificial neural networks have demonstrated human-level performance in building useful abstractions from data when tested on well-defined and constrained tasks. In computer vision, deep learning (DL) models that learn visual abstractions from raw images show strong validation performance on curated test datasets for image recognition (Krizhevsky et al., 2012), object detection (Ren et al., 2015), and scene classification (Zhou et al., 2017). However, unlike humans, DL models struggle with isolating these abstractions and systematically applying them to out of distribution test scenarios (Greff et al., 2020).

One important way in which humans both build and reason about abstractions is through analogies (Mitchell, 2021). There are several theories in human cognitive science about how humans perform analogical reasoning. Structure Mapping Theory (SMT) posits that perceptual information can be broken down into a domain consisting of objects and attributes, and structural relations between the attributes in the domain (Gentner, 1983). Consequently, SMT defines an analogy as a mapping between the structural relations across two domains, with no mapping of the attributes. For example, to draw an analogy between the solar system and an atom, we can map the relational structure (Planet *revolves around* Sun $\implies$ Electron *revolves around* Nucleus, Sun *more massive than* Planets $\implies$ Nucleus *more massive than* Electrons). However, the domain attributes, such as the Sun being *yellow* or *hot* are not mapped to the Nucleus in drawing an analogy.

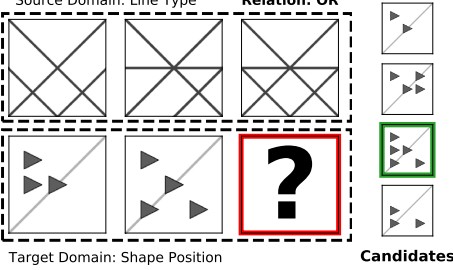

Figure 1: Abstract visual analogy problem

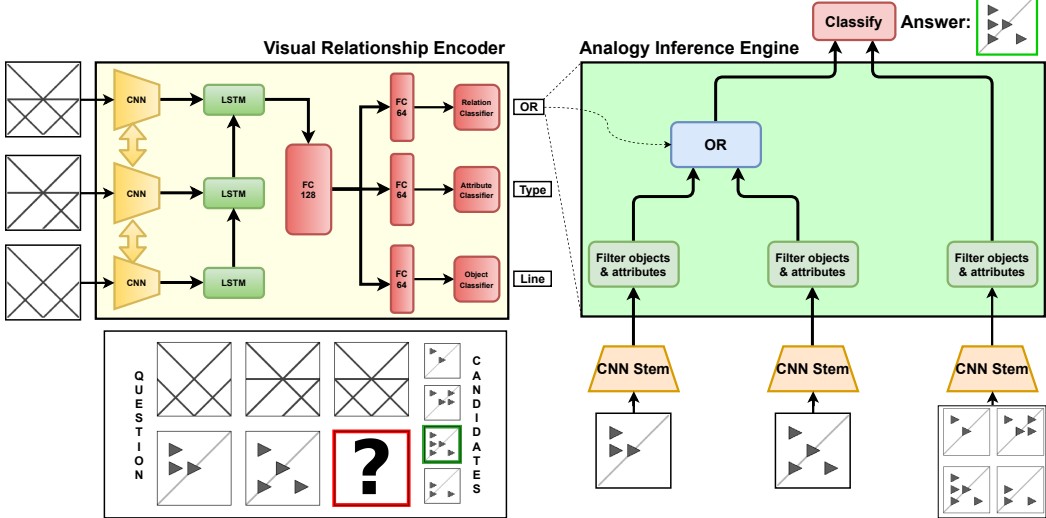

Figure 2: Neural Structure Mapping overview. Inset: RPM Analogy Problem from Fig 1. Left (yellow box): Visual Relationship Encoder to extract the object, attribute and relationship from the first row of panels. Right (green box): Analogy Inference Engine that uses the relationship label to configure a neural module net for matching the correct candidate to the second row of panels.

This idea behind structure mapping was utilized by Hill et al. (2019) as a prior in training ML models for learning analogies. They first constructed a dataset on learning abstract visual analogies from Raven's progressive matrices (RPMs) (Raven, 2000). The abstract visual analogy task is made up of two rows of five context panels, and four possible candidate panels to complete the visual analogy. The learner must understand the higher-order relation constituted by the first row of three context panels, and then choose the candidate that satisfies the same relation with the two context panels in the bottom row as shown in Figure 1. Hill et al. (2019) showed that if the candidates presented during training were curated to maximize differences in structural relations, while being perceptually similar, the models performed more accurately in learning visual analogies. Thus, the authors verfied that incorporating structural differences to maximize the importance of structure mapping in the learning procedure also works for ML models.

However, it is not feasible to build or curate datasets to always exploit a prior in the dataset to maximize structure learning. This would involve knowing a combinatorially large number of possible relational structures beforehand and manually selecting data to maximize learning these structures, a requirement that scales poorly with an increase in possible relational structures. To address this issue, we propose a two-stage Neural Structure Mapping (NSM) framework to learn abstract visual analogies (Figure 2). Our first-stage involves a visual relationship encoder that takes as input the visual context from the source domain and predicts the component visual concepts: `objects`, `attributes` and `relationship` in a multi-task manner. We then use the derived relation to dynamically build an analogy inference engine. Similar to structure mapping in humans, our engine utilizes only the relation from the source domain to pick the candidate panel that best fits the analogy. We incorporate compositional reasoning into our approach by using a neural module network (Andreas et al., 2016; Johnson et al., 2017b; Csordás et al., 2021) as our engine. Our modular neural engine dynamically adapt its structure to align (Xu et al., 2020) with the structural `relationship`, thus explicitly incorporating the structure mapping process in our NSM model.

We train and evaluate our approach on five different generalization splits of the visual analogy dataset. These splits were proposed by Hill et al. (2019) to test systematic generalization (Bahdanau et al., 2018) across object and attribute values in learning visual analogies. We show that our relationship encoder achieves a high degree of accuracy in isolating the relationship from the source domain, and, as expected, there is no performance drop across generalization splits or training regimes. For our analogy inference engine, we empirically show that it explicitly captures the structure mapping process, while generalizing systematically in learning the structural relation across visual domains.

## 2 RELATED WORK

**Learning Analogies**   Gentner (1983)'s SMT defines an analogy as a *comparison in which relational predicates, but few or no object attributes, can be mapped from base to target* domains. Symbolic models based on SMT (Falkenhainer et al., 1986), including those designed for RPMs (Lovett & Forbus, 2017), rely on first extracting a rule based representation of the domains from the perceptual input. Other cognitive analogy models based on Active Symbol theory (Hofstadter & Mitchell, 1994; Mitchell, 1993) or High-Level Perception theory (Chalmers et al., 1992) do not separate the structure extraction and mapping process, nor rely on mapping the structure syntax across domains. Our approach is not directly related to these models.

Hill et al. (2019) introduced the dataset for learning analogies from RPMs and utilized several neural network architectures to directly learn from visual data. Their key contribution was to introduce the SMT prior into the dataset during candidate selection. Webb et al. (2020) introduced Temporal Context Normalization to explicitly learn visual features that can support extrapolation, and tested it on the Visual Analogy Extrapolation Challenge (VAEC) dataset. Both of these approaches are complementary to ours. Other DL approaches most similar to ours are the Part-Composition Model (Ichien et al., 2021) and Chen et al. (2019)'s approach. However, unlike our model, these models rely on availability of very structured intermediate representations like semantic segmentation maps, and do not use the extracted structural representations to explicitly configure the reasoning model.

**Abstract Reasoning**   Hill et al. (2019)'s dataset is directly derived from the Procedurally Generated Matrices (PGM) dataset introduced by Barrett et al. (2018) to test the ability of neural networks to perform abstract reasoning on RPM problems. RPM tests are an important measure of fluid intelligence in humans (Raven, 2000), and the PGM dataset provided a sufficiently large sample size for training neural networks on this problem. Following this work, two new RPM datasets were also released: RAVEN (Zhang et al., 2019), which utilized additional rules and structured rule annotations, and V-PROM (Teney et al., 2020), which utilized real images.

DL approaches to RPM tasks can be roughly categorized into three types. The first is relation learning approaches, such as WReN (Barrett et al., 2018), which models all pairwise relationships between the matrix panels using relation networks (Santoro et al., 2017), and MXGNet (Wang et al., 2020), which learns row-based node embeddings and then performs a graph classification on the resulting candidate graphs. Second is rule learning approaches, such as DRT (Zhang et al., 2019), that learns a structured Stochastic Image Grammar of the abstract rules, and LEN (Zheng et al., 2019) that utilizes a logic embedding network along with a curriculum to learn with increasingly distracting features. Third is object-centric learning, such as Rel-AIR (Spratley et al., 2020), that obtains object embeddings generated via Attend-Infer-Repeat for each panel, and Pekar et al. (2020)'s method that utilizes both Variational AutoEncoders and an adversarial loss for candidate generation. Our model can be broadly fit into rule learning by way of the encoder, followed by relation learning via the analogy inference engine. However, our general approach towards explicitly extracting structure and mapping it is complimentary to these ideas and can be used in conjunction with several of them.

**Systematic Generalization**   Systematic generalization refers to the ability to generalize to novel concepts (out of distribution data) by understanding them as compositions of known concepts (Bahdanau et al., 2018). It is underpinned by three important characteristics: (1) systematicity, the ability to generalize to semantically related concepts; (2) productivity, the ability to iteratively apply compounding to generalize from constituent concepts to their recurrence; and (3) localism, the ability to iteratively apply reductionism to generalize from repeated constituent concepts to the singular. The ability to generalize systematically has been previously explored in human cognition (Fodor & Pylyshyn, 1988), natural language processing models (Lake & Baroni, 2018), and language-grounded computer vision models (Agrawal et al., 2017). It has its origins in human cognition and was formalized by Fodor (1975) under the 'Language of Thought' hypothesis.

Cognitive scientists have expressed reservations about the ability of connectionist models to generalize systematicitally (Fodor & Pylyshyn, 1988; Marcus, 2019). Investigation of neural language models showed that systematic generalization still poses a challenge for DL (Lake & Baroni, 2018; Loula et al., 2018). A key reason deep neural networks lack systematicity is due to their propensity towards 'shortcut learning' (Geirhos et al., 2018; 2020). This suggests that DL models rely on exploiting a few number of predictive features instead of considering all possible information about

the data while drawing conclusions. Consequently, this leads to poor generalization when the key predictive features are changed, even though the central evidence remains the same. For example, DL approaches to solving RPM problems were discovered to rely on evaluating the mode across all candidates in the RAVEN dataset and a significant drop in generalization performance was observed when the models were tested without the shortcut on the RAVEN-Fair dataset (Spratley et al., 2020). On the other hand, neural module networks (Andreas et al., 2016; Johnson et al., 2017b) generalized well systematically when their layout aligned well with specific language-grounded reasoning problems (Bahdanau et al., 2018). Hence, we utilize a modular approach in building our analogy inference engine, which in turn utilizes the `relationship` structure inferred by our encoder to automatically align itself to the problem. This is a form of conditional computation.

## 3   PROBLEM SETUP

The abstract visual analogy task (Hill et al., 2019) falls under the umbrella of Raven's Progressive Matrices (Raven, 2000) designed to test abstract reasoning and fluid intelligence in human and artificial agents. Each question consists of five context panels $P_{\text{con}}^{1-5}$, and four candidate panels $P_{\text{can}}^{1-4}$. The context panels are arranged in a matrix with two rows of three columns each, with the final panel in the second row missing. The first row of panels $P_{\text{con}}^{1-3}$ express a semantically related triplet $\mathcal{R} = \{o, a, r\}$, that is composed of two lower-order perceptual visual concepts: `object` $o$ and `attribute` $a$, and one higher-order semantic visual concept: `relationship` $r$. The objective is to choose a candidate $c \in \bigcup_{i=1}^{4}\{P_{\text{can}}^i\}$ that, upon substitution in place of the missing panel, represents the same `relationship` as the first row of panels.

The `object` ($o \in \{\text{line, shape}\}$) and `attribute` ($a \in \{\text{quantity, colour, type, size, position}\}$) together constitute the visual domain $d$ of the triplet $\mathcal{R}$. Each possible `attribute` can take ten values $v(a) \in \{1, 2, ..., 9, 10\}$ (normalized). The domain of the first row of panels $P_{\text{con}}^{1-3}$ defines the source domain $d_{\text{source}}$ of the analogy, and the domain of the second row of panels $\{P_{\text{con}}^{4-5}, c\}$ defines the target domain $d_{\text{target}}$. The dataset consists of seven unique possible domains such that $d \in \{\text{shape quantity}, \text{shape colour, shape type, shape size, shape position, line type, line colour}\}$. To test the systematic generalization in learning visual analogies, the dataset has five different generalization splits that require the ability to recognize the `relationship` across both novel domain $d$ and `attribute` $a$ values:

- **Novel Domain Transfer:** The training set consists of 42 ordered pairs of $d_{\text{source}}$ and $d_{\text{target}}$ while the remaining 7 (7*7 - 42) pairs are present only in the test set i.e.
  $d_{\text{source}}^{\text{train}} \times d_{\text{target}}^{\text{train}} \cap d_{\text{source}}^{\text{test}} \times d_{\text{target}}^{\text{test}} = \varnothing$.

- **Novel Domain Type** (`line type` **and** `shape color`)**:** The training set does not contain any problems with the held-out domain i.e. $d^{\text{train}} \notin \{\text{line type}\}$ or $d^{\text{train}} \notin \{\text{shape colour}\}$, while each problem in the test set involves the held-out domain.

- **Novel Domain Values (Interpolation and Extrapolation):** The training and test sets are made of two mutually exclusive sets of attribute values. For the extrapolation split, $v(a)^{\text{train}} \in \{1, 2, 3, 4, 5\}$ while $v(a)^{\text{test}} \in \{6, 7, 8, 9, 10\}$. For interpolation, $v(a)^{\text{train}} \in \{2, 4, 6, 8, 10\}$ while $v(a)^{\text{test}} \in \{1, 3, 5, 7, 9\}$.

Each of the splits presents a unique challenge in terms of generalization, yet shares the idea that understanding an analogy involves identifying the higher-order visual relation from the source domain and applying it to a target domain. The `relationship` ($r \in \{\text{progression, AND, OR, XOR}\}$) in the visual analogy dataset can be of classified into two types: Unary and Binary. The Unary relation, `progression`, is composed of a function applied to a single panel to produce the next panel in a row. On the other hand, the Binary relations, (`AND, OR, XOR`), are composed of a function applied to the first two panels in a row to generate the final panel. Thus, the `relationship` defines the semantics across a row of panels and is the core abstraction of analogical reasoning.

Hill et al. (2019) hypothesized that introducing a prior on the learning process that requires the learner to correctly identify the `relationship` would align with how humans learn structure for analogical mapping. They designed this prior by carefully controlling the candidate panels $P_{\text{can}}^{1-4}$ presented during model training (Figure 3). This regime, called Learning Analogies By Contrasting

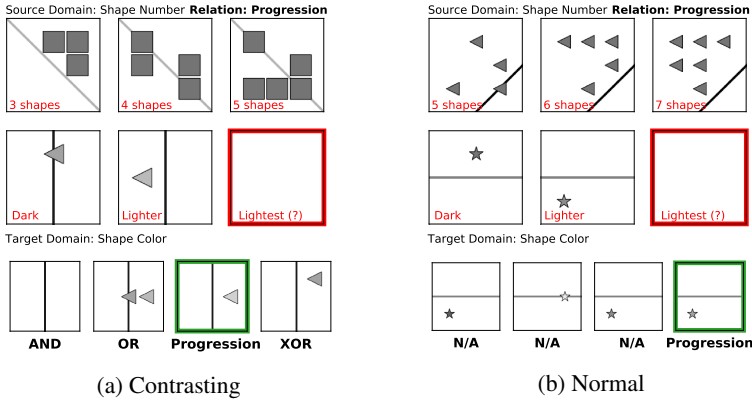

(a) Contrasting

(b) Normal

Figure 3: Two different types of candidates for the same target domain. (a) Contrasting candidates, each of which satisfies a relationship structure and requires identifying semantic structure during candidate selection. (b) Normal candidates, which are merely perceptually similar to the context.

(LABC), consists of candidates that are all semantically possible to complete the visual analogy. Each candidate satisfies one of the four possible `relationships` with the target domain panels. However, only the correct panel satisfies the `relationship` in the source domain panels. The authors found that LABC training significantly improved the ability to learn analogies in DL models compared to training with merely perceptually possible candidates (Normal training). In fact, even a mixed training regime consisting of both LABC and Normal problems showed significant improvement on the Normal training regime. To simplify presentation, in what follows we will use the term "Contrasting" for the use of such examples during training (LABC) and in evaluation.

## 4 METHOD

In our work, we take a complimentary approach to Hill et al. (2019) for learning abstract visual analogies in RPMs. Instead of depending on explicitly labeled candidates for mapping relational structure, we build a model that dynamically configures its structure based on the relational structure extracted from the source domain. Internally, our approach is made up of two different neural networks that correspond to the two separate tasks in our pipeline. The first step is the Visual Relationship Encoder, $\mathcal{R}_{\mathrm{pred}} = \phi(P_{\mathrm{con}}^{1-3})$, that predicts the visual relationship triplet $\mathcal{R} = \{o, a, r\}$ encoded in the source domain panels $P_{\mathrm{con}}^{1-3}$. Next is the Analogy Inference Engine, $c = \pi(P_{\mathrm{con}}^{4-5}, P_{\mathrm{can}}^{1-4}, r)$, that takes the predicted source relation $r$, and selects the candidate $c \in \bigcup_{i=1}^{4}\{P_{\mathrm{can}}^i\}$ that maximizes the probability of this relation in the target domain $P_{\mathrm{con}}^{4-5}$. We next provide detailed discussion on each of these networks in our pipeline. For implementation details of the networks please refer to Appendix C.

### 4.1 VISUAL RELATIONSHIP ENCODER

The Visual Relationship Encoder, $\mathcal{R}_{\mathrm{pred}} = \phi(P_{\mathrm{con}}^{1-3})$, segregates the abstract relationship triplet in the source domain panels into its constituent `object,` `attribute,` and `relationship`. This step of our NSM model corresponds to the structure extraction phase in SMT in which humans segregate the higher-order relation concepts in perceptual information from the underlying domain concepts.

The encoder $\phi$ is made up of a multi-task neural network. We use a convolutional neural network (CNN) $C_\phi$ to first extract a visual feature vector $vis_\phi^i = C_\phi(P_{\mathrm{con}}^i)$ for each context panel. The extracted visual features are then sequentially fed into a long short term memory network (LSTM) $R_\phi$ to generate visual sequence features $seq_\phi$ of the source domain. The hidden layer vector after the third sequence panel is taken as the encoded features of the source domain $seq_\phi = R_\phi(vis^{1-3})$.

The source domain feature vector $seq_\phi$ is then passed to a fully connected layer $l_\phi^{\mathrm{shared}} = FC_\phi^{\mathrm{shared}}(seq_\phi)$ that is shared between all three visual components. The shared linear fea-

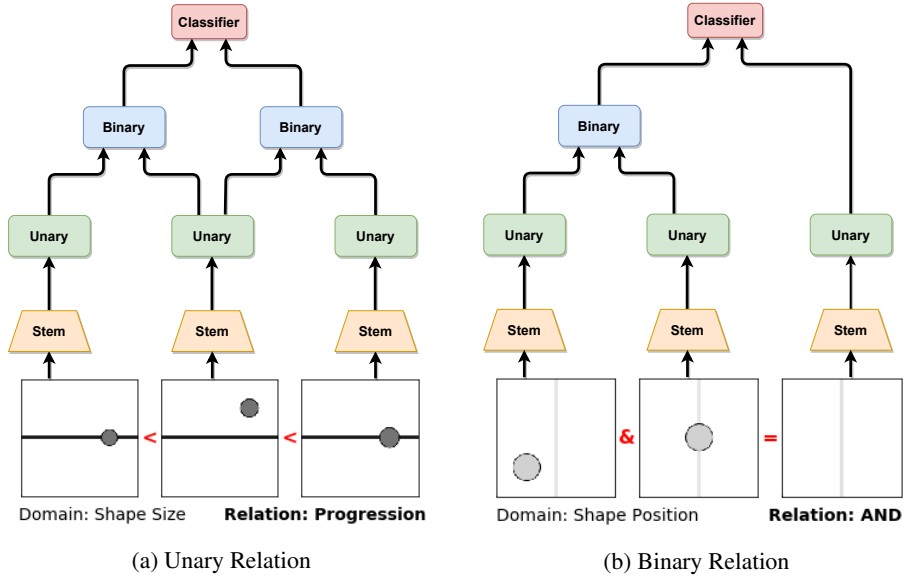

Figure 4: Types of `relationship` and corresponding Analogy Inference Engine layout. Only correct candidate panel shown. See Section 4.2 for details

ture vector $l_\phi^{\text{shared}}$ is then passed through three different fully connected layers $l_\phi^{\text{task}} = FC_\phi^{\text{task}}(l_\phi^{\text{shared}})$ for each task $t \in \{\texttt{object, attribute, relation}\}$ classification. Finally, each $l_\phi^{\text{task}}$ is passed through a fully connected layer $out_\phi^{\text{task}} = FC_\phi^{\text{out}^{\text{task}}}(l_\phi^{\text{task}})$ of sizes 2, 5, and 4 for the object, attribute, and relation prediction tasks respectively. A Softmax over the outputs $out_\phi^{\text{task}}$ yields a probability distribution over the possible $\{o, a, r\}$ values that constitute the triplet $\mathcal{R}$.

## 4.2 Analogy Inference Engine

The Analogy Inference Engine, $c = \pi(P_{\text{con}}^{4-5}, P_{\text{can}}^{1-4}, r)$, maps the source relationship $r$ extracted by the Visual Relationship Encoder to the target domain $P_{\text{con}}^{4-5}$ to identify the correct candidate $c$ that completes the visual analogy. This step of our NSM model corresponds to the structure mapping phase in SMT, in which humans apply relations extracted from the source domain to a new domain to reason about concepts in the target domain. To generalize systematically to new domains and attribute values, we utilize a modular approach (Andreas et al., 2016) to build our analogy inference engine. Similar to Johnson et al. (2017b) and Bahdanau et al. (2018), we use modules with a generic architecture. Our engine is made up of four different types of neural modules:

- The `Stem` module performs a series of convolution operations with stride=2. It takes the original grayscale images of size $1 \times 160 \times 160$ as input and returns a visual feature map of size $8 \times 9 \times 9$ ($C \times H \times W$). Each of $P_{\text{con}}^{4-5}$ and $P_{\text{can}}^{1-4}$ panels is first passed through the `Stem` module to extract visual features before being processed by the rest of the engine.

- The `Unary` module is a residual block with two $3 \times 3$ convolution layers. It receives one feature map ($C \times H \times W$) as input, and returns one feature map of ($C \times H \times W$) as output.

- The `Binary` module receives two feature maps, concatenates them along the channel dimension, and uses a $1 \times 1$ convolution to project them to $C$ dimensions. It then passes the combined feature maps through a residual block, and returns one feature map of ($C \times H \times W$) as output.

- The `Classifier` module also receives two feature maps, and projects them to $C$ dimensions. It then flattens the feature maps and passes them through two fully connected layers to generate a probability distribution over the relations.

Each candidate panel is fed parallely to the engine to generate its predictions for all four relations. In order to generate the candidate selection, we subset the probability of `relationship` $r$ ($r_{\text{pred}}$ during inference) for all candidates, and select the candidate with the highest corresponding probability.

The layout of the module network is chosen adaptively from two possible layouts based on the source domain relation at inference time. Each layout first processes the individual panels through the `Stem` module to extract visual features, followed by the `Unary` module. In the first layout, the `Unary` module outputs for the target domain panels $P_{con}^{4-5}$ are combined using a `Binary` module, and the outputs for the second panel $P_{con}^5$ and each candidate panel $P_{can}^{1-4}$ are combined using another `Binary` module (Figure 4a). The outputs of both the `Binary` modules are then fed into the `Classifier` module. In the second layout, the `Unary` module outputs for the target domain panels $P_{con}^{4-5}$ are similarly combined using a `Binary` module. However, the output of this `Binary` module and the `Unary` candidate output is directly fed into the `Classifier` module (Figure 4b). We provide further discussion on these layouts in Appendix A.

## 5 EXPERIMENTS AND RESULTS

### 5.1 VISUAL RELATIONSHIP ENCODER

**Training** The Visual Relationship Encoder $\phi$ is trained using the source domain context panels $P_{con}^{1-3}$ and the visual relationship labels $\mathcal{R} = \{o, a, r\}$. The cross-entropy loss for each task is obtained using the multi-task network's predictions and triplet labels. The total encoder loss is calculated as a weighted sum of the individual losses for object, attribute, and relation prediction:

$$\mathcal{L}_{encoder} = \alpha * \mathcal{L}_{object} + \beta * \mathcal{L}_{attribute} + \gamma * \mathcal{L}_{relationship} \tag{1}$$

Empirically, we found that the values of $\alpha = 0.5, \beta = 0.5, \gamma = 2.0$ led to higher validation accuracy. The model was trained for 100 epochs using the Adam optimizer (Kingma & Ba, 2015) with a learning rate of $1e-4$. For each generalization split, the model was trained with contrasting, normal, as well as mixed candidates. The model with the best validation accuracy was chosen for testing (the validation set consisted of a similar set of analogy candidates as the training set).

### 5.1.1 RESULTS

We test the relationship prediction accuracy of our encoder across all five generalization splits, with each possible set of analogy candidates (Contrasting, Normal, Mixed). Our relationship encoder was able to isolate the component relationship from the source domain with a high degree of accuracy (min: 82.4%, max: 86.04%) across all generalization splits and all three types of candidates (Figure 5). Furthermore, since the encoder utilizes only the source domain information in determining the relation, we observe no drop between training and test accuracy across all generalization splits.

By identifying the visual relationship in the source domain, the output of the visual relationship encoder is able to determine the layout for the subsequent analogy inference engine. Since the inference engine chooses between only two possible layouts, a baseline that always chooses the majority class layout would lead to a correct choice in 75% of problems. Hence, our encoder must

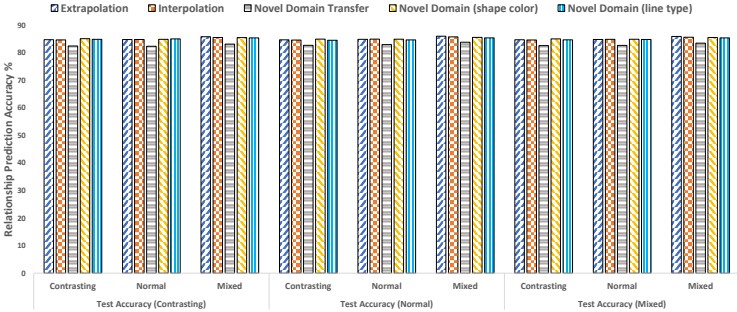

Figure 5: Relationship prediction test accuracy of our Visual Relationship Encoder. Results are first grouped (L-R) by test candidate regimes (contrasting, normal, and mixed candidates) and then sub-grouped (L-R) by training candidate regimes. Since the encoder is trained on the source domain panels, it does not have to generalize systematically and performs consistently across all splits.

achieve a high degree of accuracy in identifying the correct layout to beat this baseline. Our encoder is indeed able to identify the layout matching the reasoning problem with near perfect accuracy (min: 98.42%, max: 99.9%). The full results for layout prediction across all generalization splits and training regimes are in Table 15 in Appendix D.

## 5.2 ANALOGY INFERENCE ENGINE

**Training**  The Analogy Inference Engine $\pi$ is trained using the target domain context panels $P_{con}^{4-5}$, the candidate panels $P_{can}^{1-4}$, the ground truth candidate label, and the ground truth visual relationship label $r$. The engine was trained for 100 epochs using the Adam optimizer with a learning rate of $1e^{-4}$. Similar to the first stage, the engine was also trained with contrasting, normal, as well as mixed candidates. The model with the highest validation accuracy was chosen for testing (validation set consisted of a similar set of analogy candidates as the training set).

**Testing**  For testing, we rely on the relationship label $r_{pred}$ predicted by our encoder instead of ground truth $r$. While all baselines (see below) are trained with the full set of context panels $P_{con}^{1-5}$, our Analogy Inference Engine only utilizes the target domain panels $P_{con}^{4-5}$ for context. Hence, we also provide the results of a full context model which utilizes all context panels $P_{con}^{1-5}$. For this purpose, we leverage our encoder which is trained with the remaining context panels $P_{con}^{1} - P_{con}^{3}$. We use it to first extract the relationship $r$ from the source domain as discussed in Section 4.1. In addition, we also generate candidate probabilities from the encoder during inference, by considering each candidate in parallel followed by a Softmax over the probability of $r$ across all candidates as described in Section 4.2. The candidate probabilities of the full context NSM model are calculated as a 1:1 ensemble between the probabilities from both the inference engine and the encoder.

**Baselines**  We compare the candidate selection accuracy of our analogy inference engine with each model utilized by Hill et al. (2019). These baselines include a CNN-LSTM model, a standard ResNet50 (with 9 input channels for 9 panels), a parallel ResNet50 (with 6 channels and each candidate fed in parallel), and a Wide Relation Network. Each baseline is trained for 100 epochs using the Adam optimizer with a learning rate of $1e^{-4}$, with contrasting, normal, and mixed candidates.

### 5.2.1 RESULTS

**Does the modular analogy inference engine generalize systematically?**

We hypothesize that choosing a modular engine for analogy inference enables compositional reasoning about novel domain values (objects and attributes). In Table 2 we compare the generalization performance of our engine on the Novel Attribute Value: Extrapolation and Novel Domain: Line Type test splits of the analogy dataset. Our approach achieves the highest test accuracy across all three training candidate regimes in the Novel Domain setting when tested with contrasting candidates. Furthermore, our approach yields high test accuracy (highest or within $< 1\%$) for the Contrasting and Normal training candidate regimes in the Novel Attribute Value setting. Thus, our NSM approach is better at systematic generalization to novel visual domains than monolithic neural

| Model | Test Accuracy % (Contrasting/Normal) | | | | | |
| --- | --- | --- | --- | --- | --- | --- |
| | Novel Domain (Line Type) | | | Novel Attribute Value (Extrapolation) | | |
| | Contrasting | Normal | Mixed | Contrasting | Normal | Mixed |
| **CNN-LSTM (Hill et al., 2019)** | 76/50 | 45/57 | 75/54 | 62/45 | 43/44 | 56/39 |
| **ResNet** | 25.01/24.95 | 41.8/46.96 | 25.01/24.95 | **78.97**/55.2 | 48.5/49.55 | 62.59/52.73 |
| **ResNet-Parallel** | 79.35/**66.51** | 52.43/**76.7** | 79.67/**75.9** | 61.97/57.23 | 54.86/56.69 | 65.71/57.17 |
| **WReN** | 73.71/57.49 | 53.42/62.64 | 61.1/49.93 | 74.25/**61.36** | 61.23/**61.4** | **70.57/63.31** |
| **NSM (ours)** | 78.14/59.55 | 70.64/64.31 | 78.53/64.1 | 65.24/57.14 | 62.33/58.08 | 58.4/50.98 |
| **NSM (full context)** | **79.75**/59.55 | **76.18**/62.43 | **80.57**/65.74 | 73.36/59.94 | **66.75**/59.84 | 63.47/52.92 |

Table 1: Candidate prediction test accuracy comparison for Novel Domain and Novel Attribute values. Our approach achieves the best generalization accuracy across all models in the higher number of possible candidate scenarios (4/12), and is within $< 1\%$ of the best generalization performance in half of all scenarios (6/12). Higher accuracy == Better systematic generalization.

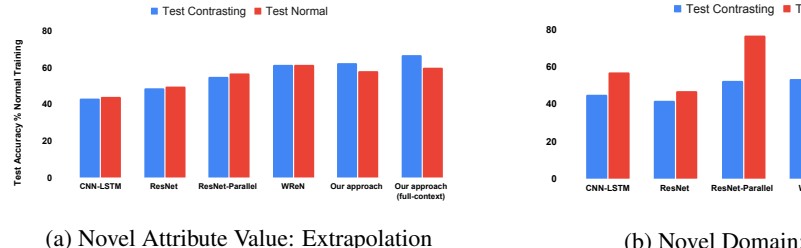

| (a) Novel Attribute Value: Extrapolation | (b) Novel Domain: Line Type |
|---|---|

Figure 6: Systematic generalization test performance with Normal training candidates which lack a prior on the dataset that maximizes structure learning during training.

networks. For completeness, we provide the results for both our evaluation setups across all training and test regimes in Tables 16 and 17 in Appendix D.2

**Does our model's prior enable it learn structure mapping?**

We specifically evaluate the Normal training scenario where the candidates presented to the learner are drawn randomly and lack the contrasting prior which explicitly promotes structure learning. Since our NSM model explicitly captures and maps the structure of the visual analogy, thus making up for the lack of a prior in the candidates, we anticipate that the test performance of our model trained with normal candidates would outperform other baselines. We visualize the generalization performance of this training scenario for the Novel Domain Value: Extrapolation and Novel Domain: Line Type generalization splits in Figure 6. We observe that our model outperforms other networks when averaged across both types of candidates during testing, demonstrating that our model's prior enables it to learns structure mapping better than other models.

**How important is structure mapping in drawing the correct analogy?**

We verify the importance of explicitly mapping structure on our NSM approach by comparing our results when the `relationship` label inferred in the first step matches the ground truth `relationship` label. We present the confusion matrix in Table 2 where it can be seen that the test candidate selection accuracy for the correct structure mapping (diagonal entries) is significantly higher than incorrect structure mapping.

|  | Prog. | XOR | OR | AND |  |  | Prog. | XOR | OR | AND |  |  | Prog. | XOR | OR | AND |
|---|---|---|---|---|---|---|---|---|---|---|---|---|---|---|---|---|
| Prog. | **90.36** | 16.72 | 22.53 | 10.88 |  | Prog. | **90.81** | 2.60 | 9.12 | 0.01 |  | Prog. | **89.93** | 30.44 | 35.57 | 21.44 |
| XOR | 20.88 | **86.49** | 57.67 | 12.32 |  | XOR | 6.61 | **89.36** | 50.04 | 1.94 |  | XOR | 35.33 | **83.58** | 65.39 | 22.83 |
| OR | 43.86 | 52.47 | **94.80** | 6.01 |  | OR | 35.69 | 52.39 | **96.19** | 0.00 |  | OR | 51.98 | 52.55 | **93.42** | 11.97 |
| AND | 5.86 | 6.18 | 2.76 | **94.72** |  | AND | 0.30 | 3.27 | 0.00 | **96.36** |  | AND | 11.51 | 9.15 | 5.57 | **93.06** |

| (a) Test Accuracy (Mixed) | (b) Test Accuracy (Contrasting) | (c) Test Accuracy (Normal) |
|---|---|---|

Table 2: Test accuracy with correct vs incorrect structure mapping. Correct Mapping == ground truth relation (rows) matches the relationship used to inform the analogy inference engine (columns).

## 6 CONCLUSIONS AND FUTURE WORK

In this work, we introduce a two-stage neural framework for learning abstract visual analogies based on the Structure Mapping Theory of human analogy making. Our first stage is a multi-task Visual Relationship Encoder that corresponds to structure extraction from perceptual information in humans. Our second stage is a modular Analogy Inference Engine that corresponds to mapping higher-order relations for structure mapping in humans. Our approach is able to generalize systematically to novel target domains, compensate for the lack of a prior in candidate selection, and successfully exploit the process of extracting and mapping the relationship structure. In future work, we plan to investigate search-based (e.g. Neural Architecture Search) as well as differentiable (e.g. attention-based) methods for generating the layouts of our Analogy Inference Engine. Furthermore, we would like explore approaches that combine our structure mapping framework into existing methods for reasoning about Raven's Progressive Matrices beyond analogy making.

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

## A How do we chose the module network layouts?

The generic architecture of our modules allows for a combinatorially large number of possible architectural layouts of our inference engine. Previous work in language-grounded reasoning tasks like VQA (Johnson et al., 2017b) or referring expression comprehension (Yu et al., 2018) uses a semantic parse tree generated from natural language to generate layouts for neural module networks. Since our inference engine $\pi(r)$ relies only on the relationship label from the source domain, it lacks the rich compositional structure available in language-grounded reasoning to arrange the modular network layout. However, generating the proper layout for neural module networks is fundamental to their ability to generalize systematically (Bahdanau et al., 2018).

As discussed in Section 4.2, we dynamically chose between two layouts of our neural module network based on the `relationship` type. Our rationale behind choosing these layouts comes from the knowledge of the possible `relationships` our engine has to reason about. Each layout aligns algorithmically (Xu et al., 2020) with a reasoning algorithm for identifying the relationship across three panels. The first layout models Algorithm 1 to identify a Unary relationship like `progression` (not to be confused with a `Unary` module) between panels 1, 2 and 3. The second layout models Algorithm 2 to identify a Binary relationship (`OR, AND, XOR`) across panels 1 and 2 in the last panel.

| **Algorithm 1** To Identify Unary Relation | **Algorithm 2** To Identify Binary Relation |
|---|---|
| **Require:** Panels $P_1, P_2, P_3$ | **Require:** Panels $P_1, P_2, P_3$ |
| 1: $D_1 \leftarrow extractObjectsAndAttributes(P_1)$ | 1: $D_1 \leftarrow extractObjectsAndAttributes(P_1)$ |
| 2: $D_2 \leftarrow extractObjectsAndAttributes(P_2)$ | 2: $D_2 \leftarrow extractObjectsAndAttributes(P_2)$ |
| 3: $D_3 \leftarrow extractObjectsAndAttributes(P_3)$ | 3: $D_3 \leftarrow extractObjectsAndAttributes(P_3)$ |
| 4: $R_{12} \leftarrow artihmeticOperation(D_1, D_2)$ | 4: $R_{12} \leftarrow binaryOperation(D_1, D_2)$ |
| 5: $R_{23} \leftarrow artihmeticOperation(D_2, D_3)$ | 5: Result $\leftarrow isConsistent(R_{12}, D_3)$ |
| 6: Result $\leftarrow isConsistent(R_{12}, R_{23})$ | 6: **return** Result |
| 7: **return** Result | |

We do not explicitly train our modules to correspond to any individual functional form. Instead, we expect the module instantiations to learn the corresponding function in the reasoning step simply by learning from the classification loss. For example, in the reasoning algorithms above, a `Unary` module can learn *extractObjectsAndAttributes()*, `Binary` module can learn *artihmeticOperation()* and *binaryOperation()*, and `Classifier` module can learn *isConsistent()* functions. This enables the modules to share parameters across relation functions (such as two different instantiations of *binaryOperation()* for `AND` and `OR`), as well as across objects and attributes. This also enables us to utilize general purpose modules which could potentially be arranged in several possible combinations depending on the reasoning problem, and can be easily extended beyond the abstract visual analogy setting of only four `relationships` to large number of possible `relationships` in real-world datasets like Scene Graph (Johnson et al., 2015) and CLEVR (Johnson et al., 2017a).

**Do algorithmically aligned engine layouts learn corresponding relations better?**

We chose two possible layouts for our modular engine based on their alignment with algorithms for identifying a Unary or Binary relation across three panels in RPM problems. We expect Layout-A (Figure 4a) to better align with Unary (`progression`) problems and Layout-B (Figure 4b) to better align with Binary (`AND, OR, XOR`) problems. In order to verify this empirically we train each layout individually with similar training hyperparameters as the full engine. We control the overall validation accuracy to select trained models which perform similarly across the full dataset. Next, we compare the test accuracy for Unary and Binary relations for the trained Layout-A and Layout-B models in Table 3.

In practice, we found significant stochasticity in the performance between Unary and Binary relation types across both layouts. From the observed results, we cannot conclusively say that our layouts necessarily align better with one or the other relationship problem. One possible explanation for this could be that there can be alternate algorithms for evaluating the same relation, which align algorithmically with these layouts. This is similar to Xu et al. (2020)'s observation that a neural network which would have not aligned with the Bellman-Ford algorithm was found to align well with a different algorithm for the same dynamic programming problem.

| Modular Layout | Training Accuracy | | | Test Accuracy % (Contrasting) | | | Test Accuracy % (Normal) | | | Test Accuracy % (Mixed) | | |
|---|---|---|---|---|---|---|---|---|---|---|---|---|
| | Contrasting | Normal | Mixed | Contrasting | Normal | Mixed | Contrasting | Normal | Mixed | Contrasting | Normal | Mixed |
| Layout A | **88.47**/94.66 | **91.81**/91.3 | 91.72/**93.91** | **87.63**/94.21 | **89.74**/90.89 | 91.51/**95.19** | **79.54**/76.32 | **90.76**/90.6 | 91.47/**92.03** | **83.53**/85.3 | **90.26**/90.74 | 91.49/**93.61** |
| Layout B | 86.44/**95** | 90.79/**92.4** | **92.85**/93.54 | 85.35/**94.28** | 77.6/90.28 | **91.96**/94.73 | 74.83/75.05 | 89.5/**91.43** | **92.83**/91.86 | 80.01/84.7 | 83.64/**90.85** | **92.4**/93.3 |

Table 3: Comparison of the analogy inference engine layouts on corresponding (unary/binary) `relationship` types

**Does having an adaptive engine enable better analogy inference?**

We use an adaptive modular engine for analogy inference which chooses between two possible module net layouts at inference time. This enables our engine to compositionally reason about the underlying relation. We expect our adaptive setup to outperform a static layout since the engine can then chose to use a better fit layout according to the mapped `relationship` structure. To verify our hypothesis, we compare our input-adaptive engine to a fixed engine. We chose each candidate layout described in Section 4.2 as a possible inference engine. We call these fixed models Engine-A and Engine-B, and train them similar to our full model. We then compare the test accuracy performance of our full model, which we label Engine-Full, against the test accuracy fixed models.

| Inference Engine | Training Accuracy % | | | Test Accuracy % (Contrasting) | | | Test Accuracy % (Normal) | | | Test Accuracy % (Mixed) | | |
|---|---|---|---|---|---|---|---|---|---|---|---|---|
| | Contrasting | Normal | Mixed | Contrasting | Normal | Mixed | Contrasting | Normal | Mixed | Contrasting | Normal | Mixed |
| Engine-A | 93.39 | 91.41 | 93.46 | 92.88 | 90.66 | 94.44 | 77.00 | 90.63 | 91.91 | 84.94 | 90.65 | 93.18 |
| Engine-B | 93.24 | 92.07 | 93.40 | 92.47 | 87.70 | 94.16 | 75.01 | 91.02 | 92.06 | 83.74 | 89.36 | 93.11 |
| Engine-Full | 93.12 | 91.04 | 93.32 | 91.07 | 90.40 | 93.30 | 74.57 | 88.92 | 90.02 | 82.81 | 89.66 | 91.65 |

Table 4: Comparison of fixed analogy inference engines versus the full adaptive engine

Empirically, we found that the adaptive engine was not able to outperform the fixed engines, contrary to our expectation. We believe there are two main reasons for this: (1) Our individual engines are not sufficiently differentiated in their performance on their respective reasoning tasks (as shown in Table 3), so there is not a significant performance gain in choosing one over the other, and (2) the number of possible reasoning scenarios is quite limited in the abstract visual analogy problem, and hence the adaptive engine is not able to benefit from learning a large number of mixture of experts.

# B VISUAL RELATIONSHIP ENCODER ABLATIONS

## B.1 MULTI-TASK VERSUS MULTI-LABEL VISUAL RELATIONSHIP ENCODER

In our two-step structure mapping approach, we treat the structure extraction problem as a multi-task learning problem where the `relationship`, `object`, `attribute` prediction are treated as separate learning tasks. We could have alternatively treated the structure extraction problem as a multi-label classification problem and instead predicted the full triplet $\mathcal{R} = \{o, a, r\}$ from our model. A multi-label problem could in principle learn from added network interactions between the final classification layer weights for the domain components.

In order to draw a comparison between both these learning approaches, we trained a multi-label visual relationship encoder which predicts the triplet $\mathcal{R} = \{o, a, r\}$ as its output. We replaced the $l_\phi^{\text{task}} = FC_\phi^{\text{task}}(l_\phi^{\text{shared}})$ layers in out multi-task encoder with a $l_\phi^{\text{shared}_2} = FC_\phi^{\text{shared}_2}(l_\phi^{\text{shared}})$ layer, and the final classifiers $out_\phi^{\text{task}} = FC_\phi^{\text{out}^{\text{task}}}(l_\phi^{\text{task}})$ of sizes 2, 5, and 4, with one classifier $out_\phi = FC_\phi^{\text{out}}(l_\phi^{\text{shared}_2})$ of size 11 (2+5+4). For training, we used exactly the same procedure as the multi-task encoder (described in Section 5.1).

We compared the performance of these models by evaluating there test accuracy across three different generalization splits: Novel Domain Transfer, Novel Domain: Line Type, and Novel Attribute Value: Extrapolation. Since our encoder is not trained with the candidate panels, we restrict ourselves to one training regime (Contrasting) for this comparison. Our observations are reported below in Table 6.

| Encoder | Extrapolation % Accuracy | | | | Novel Domain Transfer % Accuracy | | | | Novel Domain (line type) % Accuracy | | | |
|---|---|---|---|---|---|---|---|---|---|---|---|---|
| | Training | Test (Contrasting) | Test (Normal) | Test (Mixed) | Training | Test (Contrasting) | Test (Normal) | Test (Mixed) | Training | Test (Contrasting) | Test (Normal) | Test (Mixed) |
| Multi-task | 86.23 | 84.78 | 84.75 | 84.76 | 86.96 | 82.52 | 82.75 | 82.64 | 86.16 | 84.93 | 84.58 | 84.75 |
| Multi-label | 86.11 | 84.84 | 84.98 | 84.91 | 85.31 | 84.13 | 84.38 | 84.25 | 85.80 | 85.16 | 85.15 | 85.16 |

Table 5: Comparison of `relationship` prediction accuracy of a multi-task versus a multi-label visual relationship encoder

We found the performance of the two types of relationship encoders to be quite similar in terms of the test accuracy for predicting the `relationship` from the source domain panels. In fact, the multi-label encoder was slightly better than the multi-task encoder across all generalization splits. Despite this, we still went ahead with using the multi-task encoder in our two-step approach since the multi-task learning network scales much better with the number of objects, attributes, and relationships in the perceptual domain.

## B.2 MULTI-TASK VERSUS RELATIONSHIP-ONLY VISUAL RELATIONSHIP ENCODER

We also performed an ablation study on predicting only the visual `relationship` from the encoder and comparing it with the full-task of predicting `relationship`, `object`, `attribute`. We found that predicting only the relationship structure is competitive with a multi-task predictor in the Novel Domain Transfer regime, and since our encoder does not perform generalization across domains or candidate types this should hold for other regimes as well. This does not take away from our central contribution of incorporating structure mapping into our model, rather it only highlights the importance of structure extraction as the first-step of our model.

| Encoder | Novel Domain Transfer % Accuracy (Contrastive Training) | | | | Novel Domain Transfer % Accuracy (Normal Training) | | | |
|---|---|---|---|---|---|---|---|---|
| | Training | Test (Contrasting) | Test (Normal) | Test (Mixed) | Training | Test (Contrasting) | Test (Normal) | Test (Mixed) |
| Multi-task | 86.96 | 82.52 | 82.75 | 82.64 | 86.66 | 82.40 | 83.02 | 82.71 |
| Relationship-only | 84.80 | 83.61 | 83.6 | 83.61 | 85.74 | 83.68 | 83.80 | 83.74 |

Table 6: Comparison of `relationship` prediction accuracy of a multi-task versus relationship-only visual relationship encoder

## C    IMPLEMENTATION DETAILS

### C.1    VISUAL RELATIONSHIP ENCODER

| Index | Layer | Output Size |
|---|---|---|
| 1 | Panel Image Input | 1 x 160 x 160 |
| 2 | Conv(3 x 3, 1→ 8, stride 2) | 8 x 79 x 79 |
| 3 | BatchNorm | 8 x 79 x 79 |
| 4 | ReLU | 8 x 79 x 79 |
| 5 | Conv(3 x 3, 8→ 8, stride 2) | 8 x 39 x 39 |
| 6 | BatchNorm | 8 x 39 x 39 |
| 7 | ReLU | 8 x 39 x 39 |
| 8 | Conv(3 x 3, 8→ 8, stride 2) | 8 x 19 x 19 |
| 10 | BatchNorm | 8 x 19 x 19 |
| 12 | ReLU | 8 x 19 x 19 |
| 13 | Conv(3 x 3, 8→ 8, stride 2) | 8 x 9 x 9 |
| 14 | BatchNorm | 8 x 9 x 9 |
| 15 | ReLU | 8 x 9 x 9 |
| 16 | FC(8*9*9 → 128) | 128 |
| 17 | BatchNorm | 128 |
| 15 | ReLU | 128 |

Table 7: Encoder CNN architecture

| Index | Layer | Output Size |
|---|---|---|
| 1 | Encoder-CNN Output (3 panels) | 3 x 128 |
| 2 | LSTM (hidden dim = 128, sequence length = 3) | output = 3 x 128 final hidden state = 128 final cell state = 128 |

Table 8: Encoder LSTM architecture

| Index | Layer | Output Size |
|---|---|---|
| 1 | Encoder-LSTM final hidden state | 128 |
| 3 | FC(128 → 128) | 128 |
| 7 | ReLU | 128 |
| 4 | $FC_{task}(128 \to 64)$ | 64 |
| 7 | ReLU | 64 |
| 5 | $FC_{classifier_{task}}(64 \to |Task|)$ | $|Task|$ |

Table 9: Encoder Multi-Task classifier architecture. $|Task|$ = 2, 5, 4 for object, attribute, and relationship classifiers respectively

### C.2    ANALOGY INFERENCE ENGINE

| Index | Layer | Output Size |
|---|---|---|
| 1 | Panel Image Input | 1 x 160 x 160 |
| 2 | Conv(3 x 3, 1→ 8, stride 2) | 8 x 79 x 79 |
| 3 | BatchNorm | 8 x 79 x 79 |
| 4 | ReLU | 8 x 79 x 79 |
| 5 | Conv(3 x 3, 8→ 8, stride 2) | 8 x 39 x 39 |
| 6 | BatchNorm | 8 x 39 x 39 |
| 7 | ReLU | 8 x 39 x 39 |
| 8 | Conv(3 x 3, 8→ 8, stride 2) | 8 x 19 x 19 |
| 10 | BatchNorm | 8 x 19 x 19 |
| 12 | ReLU | 8 x 19 x 19 |
| 13 | Conv(3 x 3, 8→ 8, stride 2) | 8 x 9 x 9 |
| 14 | BatchNorm | 8 x 9 x 9 |
| 15 | ReLU | 8 x 9 x 9 |

Table 10: Stem module architecture

| Index | Layer | Output Size |
|---|---|---|
| 1 | Previous Module Output | 8 x 9 x 9 |
| 2 | Conv(3 x 3, 8→ 8) | 8 x 9 x 9 |
| 3 | ReLU | 8 x 9 x 9 |
| 4 | Conv(3 x 3, 8→ 8) | 8 x 9 x 9 |
| 5 | Residual: Add (1) and (4) | 8 x 9 x 9 |
| 6 | ReLU | 8 x 9 x 9 |

Table 11: Unary module architecture

| Index | Layer | Output Size |
|---|---|---|
| 1 | Previous Module Output | 8 x 9 x 9 |
| 2 | Previous Module Output | 8 x 9 x 9 |
| 3 | Concatenate (1) and (2) | 16 x 9 x 9 |
| 4 | Conv(1 x 1, 16→ 8) | 8 x 9 x 9 |
| 5 | ReLU | 8 x 9 x 9 |
| 6 | Conv(3 x 3, 8→ 8) | 8 x 9 x 9 |
| 7 | ReLU | 8 x 9 x 9 |
| 8 | Conv(3 x 3, 8→ 8) | 8 x 9 x 9 |
| 9 | Residual: Add (5) and (8) | 8 x 9 x 9 |
| 10 | ReLU | 8 x 9 x 9 |

Table 12: Binary module architecture

| Index | Layer | Output Size |
|---|---|---|
| 1 | Previous Module Output | 8 x 9 x 9 |
| 2 | Previous Module Output | 8 x 9 x 9 |
| 3 | Concatenate (1) and (2) | 16 x 9 x 9 |
| 4 | Conv(1 x 1, 16 → 8) | 8 x 9 x 9 |
| 5 | ReLU | 8 x 9 x 9 |
| 6 | FC(8*9*9 → 256) | 256 |
| 7 | ReLU | 256 |
| 8 | FC(256 → $|r|$) | $|r|$ |
| 9 | Softmax($|c|$ x $|r|$ → $|c|$) (over subset of (8) along index $r_{pred}$ for all $|c|$ candidates) | $|c|$ |

Table 13: Classifier module architecture. $|r| = 4$. $|c| = 4$.

# D  SUPPLEMENTARY RESULTS

## D.1  VISUAL RELATIONSHIP ENCODER

| GENERALIZATION SPLIT | Training Accuracy % | | | Test Accuracy % (Contrasting) | | | Test Accuracy % (Normal) | | | Test Accuracy % (Mixed) | | |
|---|---|---|---|---|---|---|---|---|---|---|---|---|
| | Contrasting | Normal | Mixed | Contrasting | Normal | Mixed | Contrasting | Normal | Mixed | Contrasting | Normal | Mixed |
| Extrapolation | 86.23 | 86.44 | 86.75 | 84.78 | 84.84 | 85.86 | 84.75 | 84.92 | 86.06 | 84.76 | 84.88 | 85.96 |
| Interpolation | 86.34 | 86.54 | 86.56 | 84.71 | 84.86 | 85.57 | 84.67 | 85.01 | 85.81 | 84.69 | 84.93 | 85.69 |
| Novel Domain Transfer | 86.96 | 86.66 | 86.30 | 82.52 | 82.40 | 83.18 | 82.75 | 83.02 | 83.88 | 82.64 | 82.71 | 83.53 |
| Novel Domain (shape color) | 86.68 | 86.48 | 86.27 | 85.16 | 84.91 | 85.55 | 85.00 | 84.98 | 85.62 | 85.08 | 84.95 | 85.58 |
| Novel Domain (line type) | 86.16 | 86.23 | 86.30 | 84.93 | 85.07 | 85.46 | 84.58 | 84.70 | 85.46 | 84.75 | 84.88 | 85.46 |

Table 14: Relationship prediction accuracy of our visual relationship encoder

| GENERALIZATION SPLIT | Training Accuracy % | | | Test Accuracy % (Contrasting) | | | Test Accuracy % (Normal) | | | Test Accuracy % (Mixed) | | |
|---|---|---|---|---|---|---|---|---|---|---|---|---|
| | Contrasting | Normal | Mixed | Contrasting | Normal | Mixed | Contrasting | Normal | Mixed | Contrasting | Normal | Mixed |
| Extrapolation | 99.86 | 99.82 | 99.98 | 99.50 | 99.46 | 99.90 | 99.49 | 99.46 | 99.88 | 99.50 | 99.46 | 99.89 |
| Interpolation | 99.92 | 99.95 | 99.98 | 99.47 | 99.58 | 99.88 | 99.41 | 99.59 | 99.85 | 99.44 | 99.59 | 99.86 |
| Novel Domain Transfer | 99.85 | 99.88 | 99.77 | 98.61 | 98.48 | 98.86 | 98.42 | 98.59 | 98.93 | 98.51 | 98.53 | 98.89 |
| Novel Domain (shape color) | 99.75 | 99.73 | 99.69 | 99.45 | 99.43 | 99.62 | 99.45 | 99.44 | 99.65 | 99.45 | 99.43 | 99.63 |
| Novel Domain (line type) | 99.96 | 99.98 | 99.99 | 99.63 | 99.66 | 99.88 | 99.65 | 99.65 | 99.89 | 99.64 | 99.65 | 99.88 |

Table 15: Engine layout prediction accuracy of our visual relationship encoder

## D.2  ANALOGY INFERENCE ENGINE

| GENERALIZATION SPLIT | Training Accuracy % | | | Test Accuracy % (Contrasting) | | | Test Accuracy % (Normal) | | | Test Accuracy % (Mixed) | | |
|---|---|---|---|---|---|---|---|---|---|---|---|---|
| | Contrasting | Normal | Mixed | Contrasting | Normal | Mixed | Contrasting | Normal | Mixed | Contrasting | Normal | Mixed |
| Extrapolation | 97.31 | 99.58 | 99.08 | 65.24 | 62.33 | 58.40 | 57.14 | 58.08 | 50.98 | 61.19 | 60.20 | 54.69 |
| Interpolation | 97.47 | 99.78 | 98.90 | 93.46 | 82.15 | 94.55 | 69.81 | 95.11 | 94.03 | 81.33 | 88.80 | 94.29 |
| Novel Domain Transfer | 97.31 | 95.70 | 96.44 | 87.96 | 87.48 | 90.81 | 73.07 | 86.78 | 87.94 | 80.50 | 87.13 | 89.38 |
| Novel Domain (shape color) | 95.82 | 97.61 | 97.75 | 77.79 | 75.64 | 82.69 | 58.72 | 66.27 | 70.26 | 68.25 | 70.96 | 76.47 |
| Novel Domain (line type) | 94.90 | 95.40 | 95.38 | 78.14 | 70.64 | 78.53 | 60.23 | 64.31 | 64.10 | 69.18 | 67.48 | 71.32 |

Table 16: Candidate prediction accuracy of our two-step model

| GENERALIZATION SPLIT | Training Accuracy % | | | Test Accuracy % (Contrasting) | | | Test Accuracy % (Normal) | | | Test Accuracy % (Mixed) | | |
|---|---|---|---|---|---|---|---|---|---|---|---|---|
| | Contrasting | Normal | Mixed | Contrasting | Normal | Mixed | Contrasting | Normal | Mixed | Contrasting | Normal | Mixed |
| Extrapolation | 96.96 | 99.41 | 98.52 | 73.56 | 66.75 | 63.47 | 59.94 | 59.84 | 52.92 | 66.65 | 63.29 | 58.19 |
| Interpolation | 97.18 | 99.68 | 98.73 | 93.11 | 85.40 | 94.95 | 70.88 | 94.86 | 93.89 | 81.71 | 90.25 | 94.41 |
| Novel Domain Transfer | 90.81 | 88.75 | 91.25 | 88.57 | 88.61 | 91.40 | 73.02 | 86.80 | 88.15 | 80.80 | 87.71 | 89.78 |
| Novel Domain (shape color) | 95.26 | 96.91 | 97.23 | 78.43 | 79.50 | 83.15 | 58.08 | 66.18 | 69.66 | 68.25 | 72.83 | 76.40 |
| Novel Domain (line type) | 94.15 | 94.58 | 94.84 | 79.75 | 76.18 | 80.57 | 59.55 | 62.43 | 65.74 | 69.65 | 69.30 | 73.15 |

Table 17: Candidate prediction accuracy of our full-context ensemble

