# OpenReview forum: "Neural Structure Mapping For Learning Abstract Visual Analogies"
_ICLR.cc/2022/Conference — ICLR 2022 Submitted_

### Official Review · Reviewer_z4sn · 2021-10-31

**Correctness:** 2
**Technical Novelty And Significance:** 2
**Empirical Novelty And Significance:** 2
**Recommendation:** 5
**Confidence:** 5

**Main Review:**

I like how the authors motivate the work in a way that can be connected to the classic structure mapping engine, as this theory is an agreed-on approach in cognition for how analogy is made. And certainly this theory could be better leveraged for the learning community to solve the task of interest in this work. I also appreciate the engineering efforts the authors make in this work that notably improve over earlier methods in this challenging task.

However, I'm particularly concerned on the novelty of this work. Using a neural modular architecture is anything but new in the reasoning community. Earlier works have extensively explored the potential of such an approach in VQA, and moreover, the NS-CL method further manages to jointly train both the visual component and the reasoning component. In this work, however, the visual component is neither jointly trained with the analogy part, nor is the analogy module any different from existing modular construction methods. What's worse, there are only two possible network layouts, a much simpler design than existing cases in VQA. Therefore, I only see this work as an application of the existing modular method in a new domain, with very little novelty. In addition, the DRT model in Zhang et. al. is exactly the modular approach, despite the fact that they call it DRT instead of neural modular network. And that model was among the very first trials in such tasks.

Besides, the authors motivate this intuitive method with SMT, which becomes no more than a story wrapper in the paper. The theory is there, but apart from *claiming* the theory and the model are related, there is not any connection between the theory and the model: in what sense is analogy inferred? The method just simply fixes the structure and selects it based on a predicted label trained from ground truth.

For the experimental evaluation, I'm not sure why the authors only select the simple models to compare. Over the years, there have been quite some new works in this topic, some have already been mentioned in the literature review, *e.g.*, MXGNet, LEN, Rel-AIR, SCL, et. al.. What about comparing with them? I also notice that the dataset used is an incomplete version the traditional Raven task. What is the model's performance on the full matrix?

Besides, if you only need the relation, why bother predicting object and attribute?

**Summary Of The Paper:**

A model called Neural Structure Mapping (NSM) is introduced to solve the task of abstract visual analogy making. The NSM model consists of a visual relationship encoder and an analogy inference engine. The visual relationship encoder extracts the visual domain elements, including object, attribute, and relation, while the analogy inference engine is a neural modular architecture that constructs the model layout based on the relation and predicts the final answer. On the dataset proposed by Hill et. al., the NSM shows better performance than other baselines.

**Summary Of The Review:**

I'm concerned on the novelty of the work and the evaluation performed.

I have read the rebuttal and other reviews. As the authors agree on most of the comments, I decide not to raise the score. But I do acknowledge that
1. The DRT may not fall into the modular network domain though the idea looks very similar.
2. I cannot quite connect the model and the structural mapping theory despite other reviewers' opinion.
3. I would still encourage comparison with other models as the tasks considered are related.

---

> ### Author Response · Authors · 2021-11-21
> **Response to Reviewer z4sn (1/2)**
>
> Response to reviewer z4sn
>
> We thank the reviewer for their feedback and for acknowledging that our paper's approach to leveraging SMT in analogical reasoning is important. We will now address their concerns in order:
>
> > concerned on the novelty of this work
>
> We acknowledge the reviewer’s concern that “using a neural modular architecture is anything but new in the reasoning community”. However, our approach does not derive its novelty from the use of neural module networks. The central contribution of our method is introducing a structure mapping prior in our model based on extracting the relationship structure from the source domain and then mapping it for reasoning in the target domain. The mapping itself is performed via informing the inference engine layout and also by determining which relationship softmax scores are used as candidate scores. This approach is similar to the cognitive Structure Mapping theory of human analogy making. We believe that this key contribution is indeed novel, but we concede that novelty is a subjective measure and respect the reviewer’s opinion.
>
> We also agree with the reviewer that our approach does not train the two models in our pipeline end-to-end. However, this is by design and not a deficiency. Our structure mapping approach is explicitly built on decoupling the structure extraction phase of analogy making from the structure mapping phase, and training our models end-to-end defeats this purpose. We have also compared our approach with other approaches which are similar to ours (CNN-LSTM and ResNet) but are trained end-to-end. Our two-step approach outperforms these models when it comes to systematic generalization as shown in our results.
>
> Regarding the DRT model being “exactly the modular approach, despite the fact that they call it DRT instead of neural modular network”, we contend that the DRT is not exactly the neural module network as proposed by Andreas et al. and commonly understood in the reasoning literature. This is indicated by the DRT authors explicitly saying that their method is based on the Tree-LSTM [1] architecture, despite being familiar with research on modular architectures as they cited [2] which uses modular networks. The key differences between our modular network and DRT are:
>
> * **Lack of neural modules:** The DRT model uses single fully-connected layers at each node in their model versus our approach of using multi-layer neural modules, which is standard in this literature.
> * **Instantiation of the nodes:**  In modular architectures, the modules are instantiated from a structured representation (text in VQA, relationship value in our approach) but they are trained without hard coding this representation in order to enable systematic generalization, and the modules independently learn the semantic meaning of the concept used for instantiation. In DRT, the nodes are instantiated using pre-trained word vector embeddings as inputs which are not trained.
>
> > there is not any connection between the theory and the model
>
> We acknowledge the reviewer’s concern regarding NSM not utilizing SMT beyond its motivation, but we do not agree with them. In fact, reviewer zpNi explicitly pointed out and praised our model’s connection to SMT: **“Interesting proposal to implement an architecture that embodies a classic theory of analogy learning (Gentner 1983)...the relationship to Gentner’s Structure Mapping Theory was clear in this component, as the Visual Relationship Encoder outputs an explicit representation of ‘structure’, which is here manifest in the relationship between separate visual scenes.”**
>
> We believe that there might be a slight miscommunication that led to the reviewer’s concern as they mention *“The method just simply fixes the structure and selects it based on a predicted label...”* The structure we refer to in our paper is the **relationship structure** obtained by our Visual Relationship Encoder, not the layout of the neural module network as is inferred from the reviewer’s comment. Furthermore, we do not ‘fix’ the layout. We dynamically choose between different possible layouts based on the inferred relationship structure at test time.
>
> Our response is continued in the next comment where we address the clarifying questions raised by the reviewer.

---

> > ### Author Response · Authors · 2021-11-21
> > **Response to Reviewer z4sn (2/2)**
> >
> >  We will now provide clarifications to the questions raised by the reviewer:
> >
> > > there have been quite some new works ... What about comparing with them? What is the model's performance on the full matrix?
> >
> > The reviewer pointed out that we do not use the full matrix-completion task as proposed in the PGM and RAVEN datasets, and instead focus on the subset task of learning visual analogies. The full matrix completion task involves other confounding forms of reasoning that limit its utility as a measure of analogical reasoning. We believe that studying and understanding the analogical reasoning problem is an important task in itself and to the audience at ICLR as demonstrated by [3] and pointed out by other reviewers:
> > **“To illustrate the effectiveness of learned analogies, the authors chose a carefully designed benchmark that... from my perspective, is sufficiently difficult as a starting-step task and a good direction to go.”**
> >
> > With regards to not comparing against other published approaches such as MXGNet, LEN, and Rel-AIR, our original reason for not including these was that these methods rely on the availability of the full matrix completion task (such as to generate row-embeddings in LEN and MXGNet) which the analogical reasoning task lacks. However, we acknowledge that these methods could be adapted for analogy making. It is out of scope for this work but we would certainly consider adapting them for comparison in the future.
> >
> > > why bother predicting object and attribute?
> >
> > We performed an ablation study and found that predicting only the relationship structure is competitive with a multi-task predictor. This does not take away from our central contribution of incorporating structure mapping into our model, rather it only highlights the importance of structure extraction as the first-step of our model. We have added the results of this ablation study to Appendix B .2 in our revision.
> >
> > ### References
> >
> > [1] K. S. Tai, R. Socher, and C. D. Manning. Improved semantic representations from tree-structured long short-term memory networks. In Proceedings of the Annual Meeting of the Association for Computational Linguistics (ACL), 2015
> >
> > [2] R. Hu, J. Andreas, M. Rohrbach, T. Darrell, and K. Saenko. Learning to reason: End-to-end module networks for visual question answering. In Proceedings of International Conference on Computer Vision (ICCV), 2017
> >
> > [3] Hill, F., Santoro, A., Barrett, D., Morcos, A., & Lillicrap, T. (2018, September). Learning to Make Analogies by Contrasting Abstract Relational Structure. In International Conference on Learning Representations.

---

### Official Review · Reviewer_QLDy · 2021-11-01

**Correctness:** 3
**Technical Novelty And Significance:** 2
**Empirical Novelty And Significance:** 2
**Recommendation:** 5
**Confidence:** 4

**Main Review:**

This paper focused on learning abstract visual analogies, which is an important problem for machine learning models to be able to generalize. To illustrate the effectiveness of learned analogies, the authors chose a carefully designed benchmark that aims at solving generalization problems under the domain of RPM problems. This benchmark is deliberately designed to challenge models' performance on the task of novel domain (novel concept), novel domain type (novel concept composition) and novel domain values (unseen values for known concepts), which from my perspective, is sufficiently difficult as a starting-step task and a good direction to go. However, I strongly feel that the authors did not distinguish their work from existing ones. Given that the benchmark, generalization tests, and learning setup (learning by contrasting) are defined in prior works, the major contribution of this paper should have come from model design and learning method, but the novelty of the proposed model is somewhat weak. First of all, the idea of leveraging object-centric representations for solving abstract reasoning tasks has been widely discussed in RPM solving tasks, which makes the visual relationship encoder conceptually not a new thing. Further, the proposed model claimed to have borrowed ideas from modular approaches for the analogy inference engine, but the unary/binary operators mentioned are designed specifically for this RPM-based task and have fixed structures (though the authors discussed dynamic structures in the appendix). This makes the proposed two structural layouts a bit weak as (1) it lacks semantic information when compared to modular networks (does not induce anything we could use for other similar tasks), and (2) it might not meet the need for more complex scenarios (e.g. real-world analogies) as a structural mapping engine. This is somewhat supported by the fact that ResNet-Parallel could be achieving similar results under certain training settings. Given these facts, the proposed model seems to be constrained on the current task and does not have strong potential as a general solution to visual analogy learning.

Here I list some points that the authors could clarify or correct me if understood them wrong during rebuttal.
(1) To find the best model that can generalize well given the current test settings, I assumed we should be looking at normal testing accuracy on each training regime since compared with contrasting candidates, the randomly sampled candidates can be more confusing. If this is the case, should we be looking for the best model that could achieve the best normal testing accuracy? (in this case, perhaps ResNet-Parallel or WReN?)
(2) It is a bit hard to see the difference between training setups and target tasks except for the obvious gap of novel domain transfer. As all visual encoders are trained given only the source domain context panels, one could imagine this task should not be too different for each generalization split. Then is there a potential reason why such a gap exists for different training setups or splits? Is it because of the joint training of later modules or distribution shift in the benchmark itself?
(3) There are minor errors on paper writing including typos and figure errors (e.g., the highlighting candidate in Figure.1 is incorrect).

**Summary Of The Paper:**

This paper targets the problem of abstract reasoning, with a special focus on the task of learning visual analogies. The authors propose a multi-stage neural network (Neural Structure Mapping, NSM) for decomposing the problem into vision relationship recognition and concept inference. They tested their model on an existing RPM (Raven's Progressive Matrices) based visual analogy benchmark that contains different systematic generalization tests and outperformed existing models. The authors made further discussion on these experimental results to support their proposals on model designs.

**Summary Of The Review:**

Although this paper works on an interesting and potentially impactful problem, the current design of the model and illustration does not fully support the claim the authors made and is somewhat incremental on technical novelty. Therefore, I recommend a rejection at this time with the hope that the authors can make this submission stronger in the next version.

Post rebuttal:
After reading the authors' response, I still feel that the current submission is somewhat incremental and needs more solid results for justifying the authors' claims. Scores are increased from 3 to 5 given the clarifications of the authors.

---

> ### Author Response · Authors · 2021-11-19
> **Response to Reviewer QLDy (1/2)**
>
> We thank the reviewer for their feedback and for acknowledging that our paper's approach to leveraging SMT in analogical reasoning is important. Also, we would like to thank them for being open to clarifications and discussion. We will now address their concerns in order:
>
> > the novelty of the proposed model is somewhat weak
>
> We agree that learning object-centric representations is an approach that has been explored previously in research on abstract reasoning, and that modular networks have been utilized in language-grounded reasoning. However, our approach does not derive its novelty from either of these individual components. The central contribution of our method is introducing a structure mapping prior in our model based on extracting the relationship structure from the source domain and then mapping it for reasoning in the target domain. The mapping itself is performed via informing the inference engine layout and also by determining which relationship softmax scores are used as candidate scores. This approach is similar to the cognitive Structure Mapping theory of human analogy making. We believe that this key contribution is indeed novel, but we concede that novelty is a subjective measure and respect the reviewer’s opinion.
>
> > the proposed model seems to be constrained
>
> The reviewer expressed a few concerns regarding our analogy inference engine. Their issues can be summarized as (1) limited flexibility in choosing engine layouts, (2) our engine is not informed by problem semantics in its design, and (3) limited utility in extending to general analogy making and reasoning. There is validity to some of these concerns. However, (2) is disproved by the dynamic selection of the engine which is in fact informed by semantic information (the structural relationship extracted from the source domain) and it does not *“lack semantic information when compared to modular networks”.*
>
> In terms of limited flexibility of our engine as highlighted in (1), we concur that the two layouts we used for the analogical reasoning problem are not as diverse as the possible layouts in a visual question answering (VQA) problem. However, this limit is a factor of the RPM analogical reasoning problem itself rather than our neural modules. In fact, the architecture of our neural modules is directly adapted from [1] where it has been shown to perform well on VQA and not be “constrained on the current task” as issue (3) suggests. In fact, if a structured representation such as Scene Image Grammar for the RAVEN dataset is available, our modules can be directly adapted to generate a combinatorially large set of possible layouts.
>
> That said, we ultimately chose to focus on the analogical reasoning dataset since the key aim of our paper was to implement structure mapping as a prior in our model and analyze its role in analogical reasoning. Studying the full matrix completion problem would have confounded other forms of reasoning and made it harder to study the ability of our model to perform analogical reasoning in isolation. Hence we limited ourselves to the analogical reasoning problem and consequently a limited number of engine layouts.
>
> Our response is continued in the next comment where we address the clarifying questions raised by the reviewer.

---

> > ### Author Response · Authors · 2021-11-19
> > **Response to Reviewer QLDy (2/2)**
> >
> > We will now provide clarifications to the questions raised by the reviewer:
> >
> > > should we be looking for the best model that could achieve the best normal testing accuracy?
> >
> > The reviewer is incorrect in their intuition that normal candidates would be harder to distinguish versus contrasting candidates. The normal candidates are not “randomly sampled” but constructed in a way that they demonstrate perceptual similarity to the target domain. As explained by [3] and also reiterated by us, contrasting candidates are harder to distinguish than normal candidates since they capture both the perceptual similarity present in normal candidates, but also semantic similarity with the target domain. This, in turn, leads to a stronger training signal for training machine learning models as they are forced to infer the underlying semantic relationship to choose the correct candidate.
> >
> > Furthermore, we argue that it would be incorrect to consider the performance on candidate selection from normal candidates as a good measure of the model’s ability to perform analogical reasoning. In the normal candidate setting, the chances that the model has merely recognized a semantic similarity without understanding the underlying relationship are higher. Consequently, the model might learn perceptual shortcuts [4] for candidate selection instead of reasoning analogically.
> >
> > > is there a potential reason why such a gap exists for different training setups?
> >
> > The reviewer expressed their confusion regarding the origin of the domain gap between the train and target tasks. We confirm that the reviewer’s intuition of the domain gap arising because of the distribution shift in the benchmark is indeed correct. We explain this in Section 3 where we discuss how the visual domains in the training set differ from the visual domains in the test set. In summary, the attribute values and domain values that are seen in the training set are different from the test set.
> >
> > Furthermore, the reviewer wondered whether the performance gap arises because of the training procedure of modules since the encoder is trained only on the source domain, which does not affect the candidate selection process beyond informing the relationship structure. We wish to highlight that there is no difference in the training procedure of our model across training candidate splits or the test generalization splits. The gap in performance is solely a function of the novel visual domains in the generalization test sets. We have addressed this in our general response and made a note of it in our rebuttal revision.
> >
> > > minor errors
> >
> > We appreciate the reviewer for bringing typos in the paper to our notice and will ensure that they are fixed. We would also like to thank them for pointing out the incorrect highlight in Figure 1: the third candidate from the top should be highlighted as it satisfies the OR relation with the target domain. We have rectified this in our revision.
> >
> > Please let us know if we have sufficiently addressed your doubts and if there are additional questions that might help you assess the paper in a more positive light. We look forward to hearing from you.
> >
> > ### References
> >
> > [1] Justin Johnson, Bharath Hariharan, Laurens Van Der Maaten, Judy Hoffman, Li Fei-Fei, C Lawrence Zitnick, and Ross Girshick. Inferring and executing programs for visual reasoning. In Proceedings of the IEEE International Conference on Computer Vision, pp. 2989–2998, 2017b.
> >
> > [2] Chi Zhang, Feng Gao, Baoxiong Jia, Yixin Zhu, and Song-Chun Zhu. Raven: A dataset for relational and analogical visual reasoning. In Proceedings of the IEEE Conference on Computer Vision and Pattern Recognition, pp. 5317–5327, 2019.
> >
> > [3] Felix Hill, Adam Santoro, David Barrett, Ari Morcos, and Timothy Lillicrap. Learning to make analogies by contrasting abstract relational structure. In International Conference on Learning Representations, 2019. URL https://openreview.net/forum?id=SylLYsCcFm.
> >
> > [4] Robert Geirhos, Jorn-Henrik Jacobsen, Claudio Michaelis, Richard Zemel, Wieland Brendel, ¨Matthias Bethge, and Felix A Wichmann. Shortcut learning in deep neural networks. Nature Machine Intelligence, 2(11):665–673, 2020.

---

> > > ### Comment · Reviewer_QLDy · 2021-11-29
> > > **Response to the authors**
> > >
> > > Thanks to the authors for their clarification on the details and modifications of this submission. However, as shared by reviewer zpNi, I still feel that the major contribution, which is the NSM model, is somewhat incremental. Therefore, I'm willing to increase my score to 5 (marginally below the acceptance threshold) while hoping that the authors could make more solid justifications of their contribution in their future revisions.

---

> > > > ### Author Response · Authors · 2021-11-29
> > > > **Response to the reviewer**
> > > >
> > > > Thank you for reconsidering your rating and providing us feedback regarding the rebuttal.

---

### Official Review · Reviewer_zpNi · 2021-11-01

**Correctness:** 3
**Technical Novelty And Significance:** 3
**Empirical Novelty And Significance:** 2
**Recommendation:** 5
**Confidence:** 4

**Main Review:**

Strengths:

Interesting proposal to implement an architecture that embodies a classic theory of analogy learning (Gentner 1983).
Visual relationship encoder architecture was clear and logical, and the relationship to Gentner’s Structure Mapping Theory was clear in this component, as the Visual Relationship Encoder outputs an explicit representation of ‘structure’, which is here manifest in the relationship between separate visual scenes.


Weaknesses:

Clarity of text, theory and justification for work:
The paper was difficult to read in parts, as some concepts were poorly explained. For example, the main paper that this work is based on seems to be somewhat misrepresented at several points in the text.  “It is not feasible to build or curate datasets to always exploit the structure mapping prior”, “their key contribution was to introduce the SMT prior into the dataset…”. “Hill et al (2019) hypothesized that introducing a prior on the learning process that requires the learning to correctly identify the relationship would align with how humans learning structure for analogical mapping”. The work by Hill et al demonstrates that when neural networks are trained with candidates that embody contrasting relational structures (akin to a curriculum), this enables the networks to learn to make analogies more effectively.  It is not clear what the authors mean by “Structure Mapping Theory prior” in these and other places in the text, but it does not give a good intuition for what this highly related piece of work contributes to the literature. When relating the current paper to other work, the authors also do not explain specifically how their architecture differs from those proposed in previous work, which is important as the architecture is the point of novelty in this paper.

The authors justify this work by making the point that in order to learn to make analogies, a researcher cannot rely solely on using class contrasts when presenting candidate options (i.e. the method proposed by Hill et al 2019), as these require knowledge of which relationship each candidate embodies, knowledge which may not always be available to a labeller. This is a fair point to make, however here the authors are proposing a new architecture that they also train and test using the contrast method illustrated by Hill et al, so it is not as though this paper proposes an alternative to class contrasts in training. In fact, the results show that the authors’ method works best in comparison to other methods precisely under the Hill et al contrast conditions. As far as I can see, there are no test splits in which the architecture proposed here performs best under Normal (not contrast) training and test conditions. The argument that clever contrast training is “not feasible” therefore seems a little misplaced.

Interestingly, in the appendix the authors show that the adaptivity of their inference engine does not enable better analogy inference than either of the static engines. It makes the reader question why the authors left this extra complexity in their design when they show it has no impact on performance, and presumably requires additional resources to train (many more weights for this extra section in the Inference Engine). The authors do not provide a clear logical reason for its inclusion.

Results:
The generalisation results appear to be somewhat cherry picked. The previously published dataset the authors use includes 5 generalisation splits, which each assess a different component of systematic generalisation. However the authors present results from just 2 of these 5 splits in the main text (Table 1 and Figure 6), and no explanation is given for why these 2 splits were the ones presented. In appendix D the authors do provide results for their method across all generalisation splits, but there is no comparison to other methods here so the values are hard to interpret. To properly and fairly communicate their results, the authors should present data (as in Table 1) for all generalisation splits and compare these fully to other work.

Figure 5 presents performance results for just the Visual Relationship Encoder, but split across all 5 different generalisation splits and also across different contrast conditions, as though each of these combinations of conditions tests a different hypothesis. However, as the encoder takes as input only the first three (source) context squares, and generalisation splits are constructed so as to isolate different relationships between the source and target squares, these many (45) bars dont tell the reader much more than what a single bar would have communicated: that the Visual Relationship Encoder predicts with about ~85% accuracy. Correspondingly the bars are all virtually identical. Presenting them all in this way implies that the Visual Relationship Encoder might be somehow performing different types of systematic generalisation, when this is really not the case.

Finally, it is nice that the authors try to determine how important the relationship output from the Visual Relationship Encoder was for performance of the Inference Engine. However the way this specific experiment was run confounds the usefulness for the Inference Engine of knowing the relation per se, with potential correlations in performance between the two network modules on each analogy example. The correct way to run this experiment would be to artificially provide the Inference Engine with all possible relationship labels (a different 'fake' relation label per run), and then evaluate whether the Inference Engine performed better when the relationship provided matched the true underlying relationship.


**Summary Of The Paper:**

This paper proposes a new architecture for learning visual analogies, based on Gentner’s Structure Mapping Theory for how humans might draw analogies. Gentner’s theory proposes representing the relationships between objects explicitly, so that this relational structure can be reused in new domains (and suggests that this commonality in structure is what permits analogies to be made between perceptually dissimilar objects). The authors propose a neural network model architecture and test it on the Raven’s Progressive Matrices dataset. The proposed architecture first splits a series of ‘source’ visual scenes into objects, attributes and the relationships between those scenes, before feeding just the relationship head into a second network. The second network then switches between two different architectures (depending on the relation fed in). The architecture in the second network (whichever is chosen) receives the ‘source’ relationship and two ‘target’ scenes before trying to predict which of a set of 4 candidate  ‘target’ scenes completes the visual analogy between source and target. The authors test their architecture on the generalisation splits in the RPM dataset and compare test accuracy results to the baseline models used by Hill et al, 2019. The authors show that their model (which builds in additional architectural structure) performs better at a subset of tests than more general architectures.

**Summary Of The Review:**

Overall,  this paper tested an interesting hypothesis based on a classic idea from cognitive science. However the results were not thoroughly analysed, and there was a  lack of clarity in the explanation of ideas that made it hard to situate this contribution of this paper in the literature.

---

> ### Author Response · Authors · 2021-11-23
> **Response to reviewer zpNi**
>
> We sincerely thank the reviewer for their extensive feedback and for praising our method as interesting and well-motivated. We also acknowledge the reviewer’s advice regarding our experiment on verifying the usefulness of structure mapping on our model’s performance as helpful since their experiment design prevents confounding the importance of structure mapping from structure extraction.
>
> We reperformed our experiments as suggested and ran the inference engine with both the correct and the incorrect relationship. Our previous observations were validated as we found that the candidate selection test accuracy of our model is significantly higher when the correct structure (ground truth relationship) is mapped vs. when the incorrect structure is mapped. Our confusion matrix for test accuracy when trained and tested with mixed candidates on the novel domain transfer regime is given below (diagonal entries represent correct mapping), and we have updated the results in Section 5.2.1 of our revision:
>
> |       | Prog. | XOR   | OR    | AND   |
> |-------|-------|-------|-------|-------|
> | Prog. | 90.36 | 16.72 | 22.53 | 10.88 |
> | XOR   | 20.88 | 86.49 | 57.67 | 12.32 |
> | OR    | 43.86 | 52.47 | 94.80 | 6.01  |
> | AND   | 5.86  | 6.18  | 2.76  | 94.72 |
>
> We will now address the reviewer’s concerns in order:
>
> > Clarity of text and theory
>
> The reviewer expressed their confusion regarding how we presented Hill et al. (2019) in our paper’s background. We would like to confirm that the reviewer’s interpretation of Hill et al. is correct. When we wrote “Structure Mapping prior”, we meant to convey that [1] introduced a prior on the learning structure by carefully selecting candidates in order to enable their models to identify structure. We have clarified this in our revision.
>
> > Reliance on contrastive training
>
> We concur with the reviewer that our method also utilizes the relationship label in training. However, unlike Hill et al., our model utilizes the class label to infer structure and does not rely on having four carefully designed candidates in the data to learn the structure. Under normal training conditions, our method performs the best with contrasting test candidates in both the Novel Domain and Novel Attribute value regimes, and near the best (59.84 vs 61.4) with normal test candidates for the Novel Attribute Values regime. Thus, our model performs best or nearly best in ¾ of the regimes with normal train/test candidates, thus validating that it is able to infer the structure better than the baselines in absence of a prior on the data. When we stated that clever contrast training is not feasible, we meant that having a dataset where this contrast is highlighted using candidate selection is not always feasible.
>
> > Extra complexity from adaptivity
>
> The authors agree that adaptivity does not enable better analogy inference vs. the static engines. However, we conjecture this is due to the limited diversity of the tasks in RPM analogical reasoning. In real world scenarios where the analogy tasks are more diverse, we expect better systematic generalization from using an adaptive engine as demonstrated in language-grounded reasoning research. We would also like to correct the reviewer as having an adaptive engine **does not add extra complexity** as our neural modules are shared across all engine layouts.
>
> > Results on various splits
>
> We agree with the reviewers that there are a total of five splits proposed in Hill et al. (2019) and we compare our models with baselines on only two of these. However, the systematic generalization split with unseen test domains are grouped as novel domain value (line type and shape color) and novel attribute values (extrapolation and interpolation). We chose one split each in novel domain value and novel attribute value in order to cover the two types of systematic generalization regimes, and also due to compute constraints. We also chose the two domains from [1] which were the hardest in terms of systematic generalization performance in terms of test accuracy. While we believe these splits are sufficiently representative of systematic generalization performance, we will add results from other baselines for all five splits in the final version of the paper.
>
> > implies that Visual Relationship Encoder might be...performing systematic generalization
>
> We did not mean to imply that the VIsual Relationship Encoder performs any form of systematic generalization. In our paper’s introduction we mention “We show that our relationship encoder achieves a high degree of accuracy in isolating the relationship from the source domain, and, **as expected**, there is no performance drop across generalization splits or training regimes.” Since the encoder is trained and tested only on the source domain, systematic generalization does not come into play nor is it implied in any way. In order to avoid this confusion, we have also highlighted this in Sec 5.1.1 and Figure 5 caption in our revision.

---

### Official Review · Reviewer_Kq5G · 2021-11-03

**Correctness:** 2
**Technical Novelty And Significance:** 3
**Empirical Novelty And Significance:** 2
**Recommendation:** 5
**Confidence:** 4

**Main Review:**

# Strengths and weaknesses
- strengths:
  - Clear explanation of motivation
  - The ability to reason with analogies seems like an important step towards intelligent systems that reason in the same way that humans do
  - The proposed method performs well against existing work and the presented baselines
- weaknesses (see detailed comments below):
  - The information available at training time may have made the task too easy, so that meaningful comparison with existing work cannot be made
  - The setup for the task seems very simple, and it doesn't seem appropriate to call it "analogy learning"
  - Claims about the effectiveness of NSM's components could be better supported by ablation studies
  - NSM achieves good performance when semantically-contrasting alternatives are not available at train time. But given the simplicity of the environment, it's unclear when collecting these alternatives would be an obstacle.

# General comments:
- Sec 4: The task as it is presented in this work seems strictly easier than the task attempted by Hill 2019 or any of the baselines.  Analogical reasoning is a hard problem, and part of that difficulty is learning to divide the world into relationship categories. In the NSM approach, this information is provided in a fully-supervised manner at train time. Building this into the method seems to make the problem much easier. Thus, a fair comparison with the baselines cannot really be made, since none of the other baselines have access to the ground truth relationship at train time.
- Sec 4: Additionally, this approach does not scale well with increasing relationship types. The number of relationships is fixed, and if you want to expand the set of relationships that NSM can handle, you must manually categorize examples according to these new relationships and manually create new neural-module layouts.
- In Table 2, it is noted that lower inference engine performance correlates with lower encoder performance. It is claimed that this shows that the encoder is an essential part of the system. However, this is not necessarily the case. It could be that there is some quality that makes the encoder fail and also makes the inference engine fail. Perhaps this corresponds with a type of relationship. For example, the encoder and inference engine could both have a hard time on relationship $X$ and an easy time on everything else. One way to check for this, would be to stratify results by relationship type. The other way to directly test this would be to run the inference engine with the correct relationship and the inference engine with the incorrect relationship and compare inference engine performance.
- I'm not entirely convinced that the system as described can be said to have "learning analogies". The domain seems very simple and far away from the type of reasoning that humans do, in which much more sensory data must be abstracted away. However, I'm aware that it is common to talk about RPM in the context of analogy learning, so I willing to concede that others may disagree.

# Clarifying questions/suggestions
- Sec 3: Is there a reason why the "extrapolation" category is mentioned here, but doesn't appear in the results in section 5?
- Sec 4: "Instead of depending on explicitly labeled candidates for mapping relational structure,"
  - Can you explain what this means? Is it very difficult to collect semantically plausible candidates? In this domain, can't the examples be generated automatically? And doesn't the proposed NSM approach also requires access to the labeled relationship for each training example anyways?
- Sec 4.1: Why is the encoder also predicting the object and the attribute? Only the predicted relationship matters for the Analogy Inference Engine, so why is multi-task learning necessary? If it is supposed to improve the prediction of the relationship, could you include an ablation study that shows this?
- Sec 4.2: Can you explain why the architecture for the Unary module and Binary module need to be different? A priori, either architecture seems like a reasonable choice. For example, I could imagine that for a unary relation, selecting the candidate involves knowing the relationship and features of the first two matrices and nothing else.
  - In Appendix A, it seems like both layouts perform equally well across all types of relationships. So why is there a need to dynamically choose between them?
- Sec 5.1 and 5.1.1 "contrasting, normal, as well as mixed candidates"
  - I think I'm missing something here. If I understand correctly, the candidate types have no bearing on the relationship encoder training? The candidates are never fed as input to the encoder, correct? So why are the results given according to candidate selection type?
- 5.2 "The candidate probabilities of the full context NSM model are calculated as a 1:1 ensemble between the probabilities from both the inference engine and the encoder"
  - This seems like an important piece of information that should probably appear earlier -- perhaps in section 4?
  - I'm not understanding how the encoder is used to select the candidates. The Visual Relationship Encoder as shown in figure 2 has no way to take the candidate as input, correct?
- Sec 5 Table 1: What numbers are being reported here? The title suggests that each cell contains the test perfomance on the contrasting data and the test performance on the normal data, separated by a slash. But the caption suggests that each cell contains the train/test performance.
- Sec 5 Table 1: "Our approach achieves the best generalization accuracy for the most number of possible train/test candidate scenarios across all models (4/12)"
  - Where does the "12" come from? Should it say 36, since there are 36 cells in the table?
- Sec 5: Do Table 1 and Figure 6 show the same information? If so, it makes sense to only keep one of them.

# small suggestions
- Figure 5: The y-axis can be scaled down
- The numbers on Figures 5 and 6 were a little too small
- If Table 1 and Figure 6 show the same information, one of the them can probably be omitted.
- pg 6. "fed parallely" --> "fed in parallel" perhaps?



**Summary Of The Paper:**

This paper tackles the problem of analogical reasoning. In particular, it presents a framework for learning the Raven Progressive Matrices (RPM) task, an abstract analogy task.

In the RPM task, a sequence of three images from a source domain are given. There is some relationship that holds for the sequence, e.g. the third image is the union of the first two. Then, given an incomplete sequence of two images from the target domain, the third image must be chosen from a list of four possible candidates.

The proposed Neural Structure Mapping (NPM) system consists of two pieces. The first piece is the Visual Relationship Encoder. Given the source sequence of images, the encoder predicts the type of relationship exhibited in the sequence. This information is passed to the second piece, the Analogy Inference Engine. The architecture of the engine is assembled dynamically, according to the predicted relationship. The assembled network takes the target sequence and the candidate matrices as input and selects the completion of the sequence from among the candidates.

The encoder is trained with the ground truth relationship labels. The engine is trained using the ground truth candidate labels.

The paper presents an experiment to test systematic generalization, in which particular attributes are held out during train time. The NSM system is found to achieve better performance.

In contrast to Hill 2019, which presents the model with semantically-contrasting alternative candidates at train time, NSM achieves good performance even when the alternative candidates are not necessarily semantically related.


**Summary Of The Review:**

The problem of learning to reason with analogies is important, and this paper makes an attempt towards doing this in a structured way, which seems like a principled move. However, as it is currently, I cannot recommend the paper for publication. The task being learned has been made too easy, so that claims about effectiveness cannot be supported. But if the authors work on a version of this study, in which the relationship types are latent/unobserved at train time, I think it would be a very promising step towards better analogical-reasoning systems.

---

> ### Author Response · Authors · 2021-11-23
> **Response to Reviewer Kq5G (1/2)**
>
> We sincerely thank the reviewer for their extensive feedback and for praising our work's motivation, the importance of the analogical reasoning problem, and the performance of our model. We also acknowledge the reviewer’s advice regarding our experiment on verifying the usefulness of structure mapping on our model’s performance. This is a helpful suggestion as their experiment design prevents confounding the importance of structure mapping from structure extraction.
>
> We reperformed our experiments as suggested by the reviewer and ran the inference engine with both the correct and the incorrect relationship. Our previous observations were validated as we found that the candidate selection test accuracy of our model is significantly higher when the correct structure (ground truth relationship) is mapped vs. when the incorrect structure is mapped. Our confusion matrix for test accuracy when trained and tested with mixed candidates on the novel domain transfer regime is given below (diagonal entries represent correct mapping), and we have updated the results in Section 5.2.1 of our revision:
>
> |       | Prog. | XOR   | OR    | AND   |
> |-------|-------|-------|-------|-------|
> | Prog. | 90.36 | 16.72 | 22.53 | 10.88 |
> | XOR   | 20.88 | 86.49 | 57.67 | 12.32 |
> | OR    | 43.86 | 52.47 | 94.80 | 6.01  |
> | AND   | 5.86  | 6.18  | 2.76  | 94.72 |
>
> We also performed an ablation for training our encoder with only the relationship label as suggested by the reviewer in one of their clarifying questions. While we saw an improvement in relationship prediction performance in multi-task training in initial training epochs, overall the relationship prediction encoder performed competitively with the multi-task encoder. This does not take away from our central contribution of incorporating structure mapping into our model, rather it only highlights the importance of structure extraction as the first-step of our model. We have added these results to Appendix B.2 in our revision.
>
>  We will now address the reviewer’s general concerns in order:
>
>
> > task easier, domain seems very simple
>
> The reviewer pointed out that the task attempted by Hill et al. that we use as our baseline does not utilize relationship labels at training time. We would like to highlight that the standard approach in abstract visual reasoning models is to train with both the candidate labels and auxiliary rule labels unlike training with only candidate labels as Hill et al. did. Our model decouples the process of rule prediction and candidate selection between the encoder and the inference engine respectively. However, we agree with the reviewer that comparing against baselines trained with the relationship labels would be fair. The reason we chose to compare with the Hill et al. (2019) models was because we wanted to highlight both the utility of our models as well as the utility of our decoupling approach vs. training with only candidate labels. We will add a comparison with baselines trained with both candidate labels and relationship labels in the final version of the paper as there wasn’t sufficient time to run these experiments during the rebuttal period.
>
> Regarding the reviewer’s concern over simplicity of the domain, we would like to point out one of the other reviewer’s comment: **“To illustrate the effectiveness of learned analogies, the authors chose a carefully designed benchmark that... from my perspective, is sufficiently difficult as a starting-step task and a good direction to go.”**
>
> > approach does not scale well
>
> We concur with the reviewer that to expand the set of relationships the model will need training samples of the new relationship examples. However, we disagree with the reviewer that this will require us to manually create new modular layouts: our relationships and layouts do not have a 1:1 mapping. In fact, we simply rely on the arity of the relationship to design our layout. Based on our current approach, we will only need one additional layout to accommodate ternary relationships, and these three layouts will be able to handle a combinatorially large number of relationships.
>
> Our response is continued in the next comment where we address the reviewer’s clarifying questions.

---

> > ### Author Response · Authors · 2021-11-23
> > **Response to Reviewer Kq5G (2/2)**
> >
> > We will now provide clarifications to the questions raised by the reviewer:
> >
> > > “extrapolation" category ... doesn't appear in the results
> >
> > Please note Table 1 where generalization performance on extrapolation is mentioned for our model and all baselines under the Novel Attribute Value setting. Appendix D also mentions the performance of our models across all splits.
> >
> > > Is it very difficult to collect semantically plausible candidates?
> >
> > We agree with the reviewer that in a procedurally generated domain like PGM it is feasible to generate carefully selected candidates to maximize structure learning. However, in real world datasets it is hard to design this prior in the dataset. This limits the contrasting approach to learning analogies in real world abstract reasoning tasks like V-PROM [1] or its utility as a component in larger reasoning tasks.
> >
> > > why is there a need to dynamically choose between them [layouts]?
> >
> > The authors agree that adaptivity does not enable better analogy inference vs. the static engines. However, we conjecture this is due to the limited diversity of the reasoning tasks in the RPM analogical reasoning task. In real world scenarios where the analogy tasks are more diverse, we expect better systematic generalization from using an adaptive engine as demonstrated in language-grounded reasoning research [2].
> >
> > > candidate types have no bearing on the relationship encoder training
> >
> > The reviewer’s intuition regarding candidate types having no bearing on the relationship encoder training is correct. For each generalization split, we reported all three candidate splits for completeness but we could have only reported one candidate split as we utilize only the source domain. We have clarified this in our rebuttal revision in the Figure 5 caption.
> >
> > > how the encoder is used to select the candidates.
> >
> > The encoder is used for candidate selection as follows: First the three source domain panels are passed to the encoder and the source relationship is extracted. Then the two target domain panels and each candidate (a total of 3 panels) is passed to the encoder one by one to generate relationship softmax probabilities for each of the four candidates. The candidate with the highest probability for the source relationship is returned as the selected candidate.
> >
> > > Sec 5 Table 1: What numbers are being reported here?
> >
> > We do indeed report the test performance with contrasting and normal candidates. We have corrected our caption for Table 1 to avoid confusion
> >
> > > Sec 5 Table 1: Where does the "12" come from?
> >
> >  12 = 2 (generalization splits) * 3 (types of training candidates) * 2 (types of testing candidates)
> >
> > > Do Table 1 and Figure 6 show the same information?
> >
> > Fig 6 does indeed show a subset of information from Table 1. We wanted to highlight the NSM results compared to baselines when trained with normal training candidates as our method’s key advantage is its ability to perform structure extraction and mapping in this setting. Hence we considered it pertinent to show these results separately in Figure 6.
> >
> > We thank the reviewer for their minor suggestions regarding presentation. We have fixed these in our rebuttal revision.
> >
> > ### References
> >
> > [1] Teney, Damien, Peng Wang, Jiewei Cao, Lingqiao Liu, Chunhua Shen, and Anton van den Hengel. "V-PROM: A benchmark for visual reasoning using visual progressive matrices." In Proceedings of the AAAI Conference on Artificial Intelligence, vol. 34, no. 07, pp. 12071-12078. 2020.
> >
> > [2] Justin Johnson, Bharath Hariharan, Laurens Van Der Maaten, Judy Hoffman, Li Fei-Fei, C Lawrence Zitnick, and Ross Girshick. Inferring and executing programs for visual reasoning. In Proceedings of the IEEE International Conference on Computer Vision, pp. 2989–2998, 2017b.

---

> > > ### Comment · Reviewer_Kq5G · 2021-11-29
> > > **Response to author comments**
> > >
> > > I thank the authors for their clear and thorough response. However, my recommendation remains the same: marginally below the acceptance threshold.
> > >
> > > I share the same question as reviewer z4sn, who asks "...in what sense is analogy inferred?". In my view, this question is not completely answered in the work or the author responses. As part of their response to z4sn, the authors cite the dynamic adaptivity of their systems. However, in their response to my review they say:
> > >
> > > > [dynamic] adaptivity does not enable better analogy inference vs. the static engines. However, we conjecture this is due to the limited diversity of the reasoning tasks in the RPM analogical reasoning task.
> > >
> > > Dynamic adaptivity seems to be a main component of the presented system and the fact that its effectiveness cannot be borne out by experiment is concerning, since it indicates that either dynamic adaptivity is not needed or that the chosen experiment setting is inappropriate.
> > >
> > > > We would like to highlight that the standard approach in abstract visual reasoning models is to train with both the candidate labels and auxiliary rule labels
> > >
> > > I disagree that this is the correct approach in this case. If the goal is to build a system that learns analogies, then the abstract relationships to be discovered should not be provided at train time. Since the relationship label is provided at training, it's unclear how the task differs from a simple classification. At the very least, given that this work follows and builds on the results of Hill et al. 2019, I would expect that the authors would maintain the same level of task difficulty in this work.
> > >
> > > Finally, it seems that many of the design choices introduce complexity where it is not needed. As noted above, dynamic adaptivity does not matter much to performance. And, for the multi-task predictor head on the encoder, the authors also note that:
> > > > predicting only the relationship structure is competitive with a multi-task predictor.
> > >
> > > The authors do a very good job of motivating their work with a well-written history of analogy learning systems. I would encourage them to re-work this submission into one that 1) uses the same difficult setting of Hill et al. 2019 and 2) provides clear evidence that supports their design choices, e.g., dynamic adaptivity and multi-task prediction.

---

### Official Review · Reviewer_opcn · 2021-11-04

**Correctness:** 2
**Technical Novelty And Significance:** 1
**Empirical Novelty And Significance:** 1
**Recommendation:** 3
**Confidence:** 5

**Main Review:**

- The authors clearly know a lot about the literature about analogical reasoning and RAVEN. This paper provides an excellent review of related work in a concise manner.

- Technical details are largely missing. There's even not a single equation in the entire paper. This makes it difficult to believe that the proposed method would work as the authors claimed. And if it does, it poses additional questions on how general it is to other analogical problems in general.

- Results are weak. It's hard to see if the proposed method is indeed better than prior work in terms of performance. For instance, under the extrapolation setting with mixed data, the WReN is much better than the proposed one. This should be the best evidence to see if the proposed method is better.

- Figures are very difficult to understand. For instance, there's no explanation of what "filter" means in Figure 1. Bar plots in Figure 6 should have standard diveration or some other metrics in addition to the bar.

**Summary Of The Paper:**

Motivated by the structure mapping theory, this paper proposes an ad-hoc solution to address the classic RAVEN problem. The authors claim the proposed method is more general than previous methods while achieving good performance.

**Summary Of The Review:**

Good literature review and motivation, but difficult to tease out the technical contributions with weak and ambiguous results.

---

> ### Author Response · Authors · 2021-11-19
> **Response to Reviewer opcn**
>
> We thank the reviewer for their feedback and appreciation of our paper’s motivation and literature survey. We will now address their concerns in order:
>
> > Technical details are largely missing
>
> We have provided all the technical details (dataset splits, model architecture and hyper-parameters, training methodology and experimental hyper-parameters) required to perform and replicate our experiments in the main paper and Appendix C. We will also do a code release if our paper is accepted. We disagree with the reviewer’s suggestion that the lack of equations in the paper is representative of a lack of technical rigor. We politely ask the reviewer to let us know if there are specific equations or further details that we can provide which will help them better assess our paper.
>
> > Results are weak
>
> We acknowledge the reviewer’s concern that there are indeed out-of-distribution generalization distributions (like training with mixed candidates and testing on extrapolated attribute values) where our model does not outperform the baselines. We would also like to point out that we never claim that our model performs the best for all splits.
>
> However, there are several training and test splits where our model outperforms the baselines as shown in Table 1. Specifically, our model demonstrates the best generalization performance for all three types of training candidate sets for the novel domain setting, and also for the normal training candidate set for the novel attribute value setting. We feel that this claim is unsubstantiated, especially as reviewer z4sn explicitly praised our model’s design and performance: **“I also appreciate the engineering efforts the authors make in this work that notably improve over earlier methods in this challenging task.”** and reviewer Kq5G also highlighted our model’s performance  **“The proposed method performs well against existing work and the presented baselines”.**
>
> Furthermore, we also do not agree with the claim that *“this should be the best evidence to see if the proposed method is better.”* The key contribution of our paper is the structure mapping inductive bias in our model, which seeks to alleviate a lack of contrastive structural priors in the dataset in training with normal candidates. Our model demonstrates strong performance across the board in this setting as shown in Figure 6.
>
> > Figures are very difficult to understand
> * Figure 1 does not have a “filter” in the text or captions so we did not know how to interpret this comment. We ask the reviewer to please clarify their comment.
> * We will add bootstrap estimates to all bar charts to add a measure of deviation in the final version of the paper.
>
>
> We think that some of the reviewer’s concerns are misplaced and we would like to politely request them to reconsider them in light of our response. In summary, the brevity of the review and the high-level nature of the critique such as the comment about lack of equations gives us concern. We are open to discussion and would be happy to answer any additional questions. We would also be happy to provide any particular technical details that will help the reviewer better assess the paper.

---

### Author Response · Authors · 2021-11-23
**General response to reveiwers and chairs**

We thank the reviewers for their insightful feedback and comments on our paper. The reviewers praised our paper’s clear explanations, our method’s importance and relevance, and its performance in comparison to baselines. In this general response, we will address some of the concerns shared across reviewers and summarize the changes made in our rebuttal revision.

**Novelty of our work:** The central contribution of our method is introducing a structure mapping prior in our model based on extracting the relationship structure from the source domain and then mapping it for reasoning in the target domain. The mapping itself is performed via informing the inference engine layout and also by determining which relationship softmax scores are used as candidate scores. This approach is similar to the cognitive Structure Mapping theory of human analogy making and indeed novel.


**Relevance of adaptivity:** We found in our ablations that adaptivity of the inference engine does not conclusively enable better analogy making compared to static engines. However, we conjecture this is due to the limited diversity of the reasoning tasks in the Raven style analogical reasoning task. In real world scenarios where the analogy tasks are more diverse, we expect better systematic generalization from using an adaptive engine as demonstrated in language-grounded reasoning research. We would also like to point out that our neural modules are shared across all engine layouts, and as such training multiple layouts does not add to the training parameters or training complexity.


**Scope of generalization for Visual Relationship Encoder:** Our visual relationship encoder is trained only on the source domain panels, and as such its performance across various training and test splits is not indicative of systematic generalization. We had mentioned this previously in our paper (*“We show that our relationship encoder achieves a high degree of accuracy in isolating the relationship from the source domain, and, **as expected**, there is no performance drop across generalization splits or training regimes.”*) but we have reiterated this in Section 5.1 and Figure 5 in our revision.


**Evaluating importance of structure mapping:** We reperfromed our experiments to verify the importance of structure mapping to our engine’s ability to perform analogical reasoning. Two separate reviewers had noted that our previous experiment could have confounded this result with the importance of structure extraction. Hence, we ran the inference engine with all four possible relationship structures and compared the performance when the correct relationship structure was mapped to the cases when the relationship structure was mapped incorrectly. We confirmed that the candidate selection accuracy was significantly higher with the correct structure mapping, and we have included these results in Section 5.2.1 of our revision.


**Role of multi-task encoder:** We also performed an ablation for training our encoder with only the relationship label as suggested by two separate reviewers. While we saw an improvement in relationship prediction performance in multi-task training in initial training epochs, overall the relationship prediction encoder performed competitively with the multi-task encoder. This does not take away from our central contribution of incorporating structure mapping into our model, rather it only highlights the importance of structure extraction as the first-step of our model. We have added these results to Appendix B.2 in our revision.

### Summary of changes in rebuttal revision

* New experiments performed to verify the importance of structure mapping as explained before in Section 5.2.1
* New ablation to compare multi-task versus relationship-only encoder in Appendix B.2
* Table 15 in Appendix D was updated with correct numbers
* Clarified the role of the prior on the dataset as proposed by Hill et al instead of using the term ‘structure mapping prior’ in Section 1, Section 5.2.1, Section 6, and Figure 6 caption
* Figure 1 highlight of candidate was updated to reflect the correct candidate
* Figure 5 caption was updated to reiterate that encoder does not perform systematic generalization
* Table 1 caption was changed to avoid any confusion and to clarify that test accuracy is being reported
* Figure 5 y-axis was scaled down
* Figure 6 y-axis ticks were made larger

Thank you for your consideration!

---

### Decision · Program_Chairs · 2022-01-20

**Decision:**

Reject

**Comment:**

This manuscript presents an approach to handling abstract visual analogy tasks where panels of drawings are shown with a missing entry. One of a number of candidate drawings must be chosen to complete the panel. Reviewers brought up several concerns:

1. The task performed was made considerably easier by providing additional annotations at training time. This was not the case in the original task in prior work that the manuscript builds on. No convincing explanation was provided as to why this change is critical to accommodate the manuscript's contributions.

2. A key feature of the approach, the adaptive modular design, does not seem to contribute much. The authors rightly point out this may be a limitation of current benchmarks. Reviewers were sympathetic to this view but that leaves the manuscript in a tough spot: a central contribution cannot be evaluated. What is even worrisome is that without evaluating the effectiveness of the adaptive design we cannot know if it is working at all. What if adaptivity is required for some future analogy tasks but it turns out that this approach, despite seeming to be adaptive, falls short?

3. Another contribution, the multi-task encoder does not seem to provide much value in ablation experiments. The manuscript would be improved if this feature was removed or its usefulness was demonstrated.

A number of smaller issues were also brought up by reviewers.

Throughout the responses to reviewers the authors highlight that their central contribution is incorporating a structure mapping prior "The central contribution of our method is introducing a structure mapping prior ...". I would like to draw the author's attention to the fact that they had to remind 3 of the 4 reviewers to focus on this rather than another aspect of the work. That clearly indicates that the manuscript and work needs a shift in focus. I would suggest that authors double down on their structural mapping prior, eliminate all other features which turned out to be controversial or impossible to evaluate, and demonstrate the utility of their idea in two domains, i.e., including another domain. This would really highlight the core contribution.

Unfortunately, what may turn out to be a good idea, the structural mapping prior, is lost among many other complexities. I hope the authors are not discouraged and that we see this line of work again in the future.